# *Drosophila* HCN mediates gustatory homeostasis by preserving sensillar transepithelial potential in sweet environments

MinHyuk Lee[1,2,3], Se Hoon Park[4], Kyeung Min Joo[2], Jae Young Kwon[3], Kyung-Hoon Lee[2], KyeongJin Kang[1]*

[1]Neurovascular Unit Research Group, Korea Brain Research Institute, Daegu, Republic of Korea; [2]Department of Anatomy and Cell Biology, Sungkyunkwan University School of Medicine, Samsung Biomedical Research Institute, Samsung Medical Center, Suwon, Republic of Korea; [3]Department of Biological Sciences, Sungkyunkwan University, Suwon, Republic of Korea; [4]Department of Brain Sciences, DGIST, Daegu, Republic of Korea

**\*For correspondence:**
kangkj@kbri.re.kr

**Competing interest:** The authors declare that no competing interests exist.

**Abstract** Establishing transepithelial ion disparities is crucial for sensory functions in animals. In insect sensory organs called sensilla, a transepithelial potential, known as the sensillum potential (SP), arises through active ion transport across accessory cells, sensitizing receptor neurons such as mechanoreceptors and chemoreceptors. Because multiple receptor neurons are often co-housed in a sensillum and share SP, niche-prevalent overstimulation of single sensory neurons can compromise neighboring receptors by depleting SP. However, how such potential depletion is prevented to maintain sensory homeostasis remains unknown. Here, we find that the *Ih*-encoded hyperpolarization-activated cyclic nucleotide-gated (HCN) channel bolsters the activity of bitter-sensing gustatory receptor neurons (bGRNs), albeit acting in sweet-sensing GRNs (sGRNs). For this task, HCN maintains SP despite prolonged sGRN stimulation induced by the diet mimicking their sweet feeding niche, such as overripe fruit. We present evidence that *Ih*-dependent demarcation of sGRN excitability is implemented to throttle SP consumption, which may have facilitated adaptation to a sweetness-dominated environment. Thus, HCN expressed in sGRNs serves as a key component of a simple yet versatile peripheral coding that regulates bitterness for optimal food intake in two contrasting ways: sweet-resilient preservation of bitter aversion and the previously reported sweet-dependent suppression of bitter taste.

## eLife assessment

This study provides **important** new insight into how non-synaptic interactions affect the activity of adjacent gustatory neurons housed within the same sensillum. The conclusions are supported by **convincing** electrophysiological, behavioral, and genetic data. This work will be of interest to neuroscientists studying chemosensory processing or regulation of neuronal excitability.

## Introduction

Glia-like support cells exhibit close physical association with sensory receptor neurons, and conduct active transcellular ion transport, which is important for the operation of sensory systems (*Ray and Singhvi, 2021*). In mammals, retinal pigment epithelial (RPE) cells have a polarized distribution of ion

channels and transporters. They provide an ionic environment in the extracellular space apposing photoreceptors to aid their light sensing (*Sparrrow et al., 2010*). Likewise, knockdown of *Drosophila* genes encoding the Na$^+$/K$^+$ pump or a K$^+$ channel in the supporting glial cells attenuates photoreceptors (*Charlton-Perkins et al., 2017*). In addition to creating an optimal micro-environment, transepithelial potential differences (TEPs) are often generated to promote the functions of sensory organs. For example, the active K$^+$ transport from the perilymph to the endolymph across support cells in the mammalian auditory system (*Nin et al., 2008*) generates high driving forces that enhance the sensitivity of hair cells by increasing K$^+$ and Ca$^{2+}$ influx through force-gated channels. Similar designs have been found in insect mechanosensory (*Tuthill and Wilson, 2016*; *Erler and Thurm, 1981*) and chemosensory organs (*Sollai et al., 2008*; *Vermeulen and Rospars, 2004*), providing models to study physiological principles and components of TEP function and regulation. Many insect sensory receptor neurons are housed in a cuticular sensory organ called the sensillum. Tight junctions between support cells separate the externally facing sensillar lymph from the internal body fluid known as hemolymph (*Shanbhag et al., 2001*). The active concentration of K$^+$ in the dendritic sensillar lymph produces positive sensillum potentials (SP, +30–40 mV) as TEPs, which are known to sensitize sensory reception in mechanosensation (*Grünert and Gnatzy, 1987*) and chemosensation (*Gödde and Krefting, 1989*; *Syed and Leal, 2008*).

Excitation of sensory neurons drains SP, accompanied by slow adaptation of the excited receptor neurons (*Erler and Thurm, 1981*; *Syed and Leal, 2008*). This suggests that immoderate activation of a single sensory neuron can deplete SP, which decreases the activities of neurons that utilize the potential for excitation. Each sensillum for mechanosensation and chemosensation houses multiple receptor neurons (*Ray and Singhvi, 2021*). Therefore, overconsumption of SP by a single cell could affect the rest of the receptor neurons in the same sensillum, because the receptor neurons share the sensillum lymph. Indeed, the reduction of SP was proposed to have a negative effect on receptor neurons that are immersed in the same sensillar lymph; a dynamic lateral inhibition between olfactory receptor neurons (ORNs) occurs through 'ephaptic interaction,' where SP consumption by activation of one neuron was proposed to result in hyperpolarization of an adjacent neuron, reducing its response to odorants (*Zhang et al., 2019*; *Van der Goes van Naters, 2013*). As expected with this SP-centered model, ephaptic inhibition was reported to be mutual between *Drosophila* ORNs (*Zhang et al., 2019*; *Su et al., 2012*), again because the ORNs are under the influence of a common extracellular fluid, the sensillar lymph. Such reciprocal cancellation between concomitantly excited ORNs may encode olfactory valence (*Wu et al., 2022*) rather than lead to signal attenuation of two olfactory inputs. Furthermore, depending on neuron size, the lateral inhibition between ORNs can be asymmetric, albeit yet to be bilateral; larger ORNs are more inhibitory than smaller ones (*Zhang et al., 2019*). The size dependence was suggested to be due to the differential ability of ORNs to sink SP (referred to as local field potential in the study, *Zhang et al., 2019*), probably because larger cells have more membrane surface area and cell volume to move ions to or from the sensillar lymph.

Interestingly, gustatory ephaptic inhibition was recently found to be under a genetic, but not size-aided, regulation to promote sweetness-dependent suppression of bitterness (*Lee et al., 2023*). This is accomplished by blocking one direction of ephaptic inhibition. The hyperpolarization-activated cation current in sGRNs through the *Ih*-encoded hyperpolarization-activated cyclic nucleotide-gated (HCN) is necessary to resist the inhibition of sGRNs laterally induced by bGRN activation. Furthermore, such unilateral ephaptic inhibition is achieved against cell size gradient (*Lee et al., 2023*). Larger bGRNs are readily suppressed by the activation of smaller sGRNs, but not vice versa. Thus, HCN is implemented to inhibit bGRNs in terms of unilateral ephaptic inhibition when a bitter chemical is concomitantly presented with strong sweetness. Here, in addition to the ephaptic interaction, we find that the same HCN expressed in sGRNs promotes the activity of bGRNs as a means of homeostatic sensory adaptation, for which HCN prevents sGRNs from depleting SP even with the long-term exposure to the sweet-rich environment.

## Results

### HCN expressed in sweet-sensing GRNs is required for normal bitter GRN responses

The hair-like gustatory sensilla in the *Drosophila* labellum are categorized into L-, i-, and s-type based on their relative bristle lengths. Each sensillum contains 2 (i-type) or 4 (s- and L-type) GRNs along with a mechanosensory neuron. The i- and s-type bristle sensilla contain both an sGRN and a bGRN, while each L-type bristle sensillum contains an sGRN but no bGRN (*Ishimoto and Tanimura, 2004*; *Tanimura et al., 2009*; *Fujii et al., 2015*; *Weiss et al., 2011*). As a model of gustatory homeostasis, we mainly examined the i-type bristles using single sensillum extracellular recording (*Hodgson et al., 1955*; *Du et al., 2019*; *Du et al., 2016*) because of their simple neuronal composition. Compared to WT ($w^{1118}$ in a Canton S background), we observed reduced spiking responses to 2 mM caffeine in two strong loss-of-function alleles of the HCN gene, $Ih^{f03355}$ (*Fernandez-Chiappe et al., 2021*; *Hu et al., 2015*) and $Ih^{MI03196-TG4.0}/+$ (*Ih-TG4.0/+*) (*Lee et al., 2023*; *Lee et al., 2018*; *Figure 1A*). Note that *Ih-TG4.0* is homozygous lethal (*Lee et al., 2023*). A copy of the *Ih*-containing genomic fragment {*Ih*} rescued the spiking defect in $Ih^{f03355}$. The GRN responses to 50 mM sucrose were not altered in *Ih* mutants (*Figure 1B*). Other bitter chemical compounds, berberine, lobeline, theophylline, and umbelliferone, also required *Ih* for normal bGRN responses (*Figure 1—figure supplement 1*). Although we observe here that *Ih* pertains to bGRN excitability, *Ih* was previously found to be expressed in sGRNs but not bGRNs (*Lee et al., 2023*). To test whether HCN expression in sGRNs is required for bGRN activity, GRN-specific RNAi knockdown of *Ih* was performed with either *Gr64f-* (*Dahanukar et al., 2007)* or *Gr89a-Gal4* (*Weiss et al., 2011*). *Ih* knockdown in sGRNs (*Gr64f-Gal4*), but not bGRNs (*Gr89a-Gal4*), led to reduced bGRN responses to caffeine (*Figure 1C*), indicating that HCN acts in sGRNs for a normal bGRN response. Unlike the results in *Ih* mutant alleles, the spiking response of *Ih*-knock-downed sGRNs (*Gr64f* cells) to 50 mM sucrose was increased (*Figure 1D*). To exclude the possibility that *Ih* is required for normal gustatory development, we temporally controlled *Ih* RNAi knockdown to occur only in adulthood, which produced similar results (*Figure 1—figure supplement 2*). The differential effects of gene disruptions and RNAi on sGRN activity will be discussed further below with additional results. Introduction of *Ih-RF* cDNA (FlyBase id: FBtr0290109), which previously rescued *Ih* deficiency in other contexts (*Lee et al., 2023*; *Hu et al., 2015*), to sGRNs but not bGRNs restored the decreased spiking response to 2 mM caffeine in $Ih^{f03355}$, corroborating that sGRNs are required to express *Ih* for bGRN regulation (*Figure 1E*). Interestingly, ectopic cDNA expression in bGRNs of $Ih^{f03355}$ but not in sGRNs increased the spiking response to 50 mM sucrose compared to its controls (*Figure 1F*), although the same misexpression failed to raise the spiking to 2 mM caffeine. These results suggest not only that *Ih* innately expressed in sGRNs is necessary for the activity of bGRNs, but also that *Ih* expression in one GRN may promote the activity of the other adjacent GRN in *Ih*-deficient animals.

### Loss of *Ih* in sGRNs reduced the sensillum potential in the gustatory bristle sensilla

We speculated that the *Ih*-dependent lateral boosting across GRNs might involve a functional link between GRNs. Such a physiological component could be SP, since the sensillar lymph is shared by all GRNs in the sensillum and SP sets the spiking sensitivity (*Tuthill and Wilson, 2016*). SP is known as a transepithelial potential between the sensillum lymph and the hemolymph, generated by active ion transport through support cells (*Figure 2A*, Left). To measure SP, we repurposed the Tasteprobe pre-amplifier to record potential changes in a direct current (DC) mode (see Materials and methods for details), which was originally devised to register action potentials from sensory neurons. With the new setting, the contact of the recording electrode with a labellar bristle induced a rise in potential (*Figure 2A*, Right). The recording was stabilized within 20 s, and a raw potential value was acquired as an average of the data between the time points, 20 and 60 s after the initial contact (*Figure 2A*). After the examination of all the bristle sensilla of interest, the fly was impaled at the head to obtain the DC bias (also known as DC offset), which insects are known to exhibit in the body independent of SP (*Marion-Poll and van der Pers, 1996*; *Figure 2B*). To examine whether the DC bias varies at different body sites, we surveyed the DC bias at four different locations of individual animals, the abdomen, thorax, eye, and head. This effort resulted in largely invariable DC bias readings (*Figure 2B and C*).

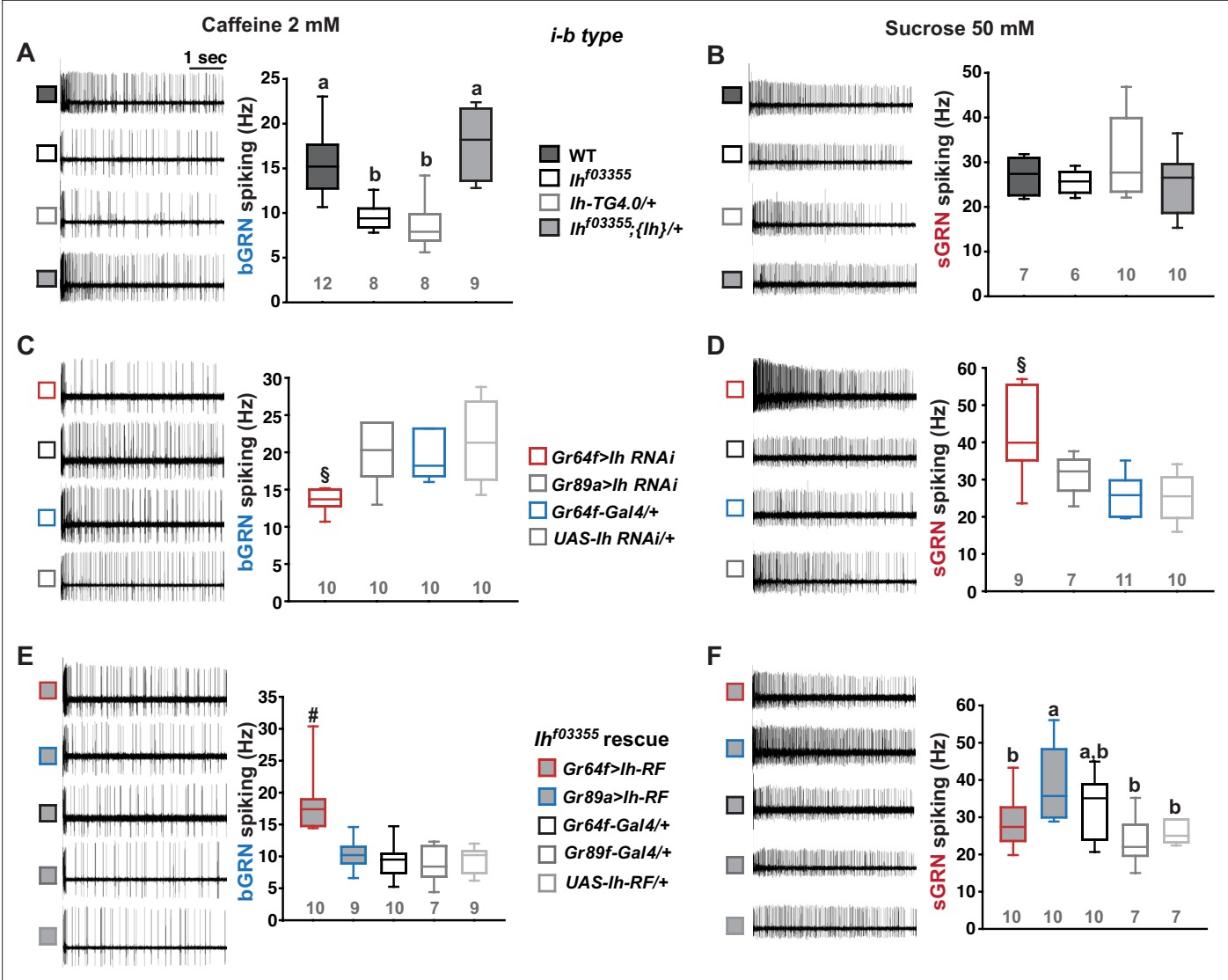

**Figure 1.** Hyperpolarization-activated cyclic nucleotide-gated (HCN) channel is necessary for the normal activity of bitter-sensing GRNs (bGRNs), although expressed in sweet-sensing GRNs (sGRNs). Representative 5 s-long traces of sensillum recording with either caffeine or sucrose at the indicated concentrations, shown along with box plots of spiking frequencies. (A) Caffeine-evoked bitter spiking responses of wild-type (WT), the $Ih$-deficient mutants, $Ih^{f03355}$ and $Ih$-TG4.0/+, and the genomic rescue, $Ih^{f03355};\{Ih\}/+$. (B) Sucrose responses were similar among the genotypes tested in (A). (C) $Ih$ RNAi knockdown in sGRNs, but not bGRNs, reduced the bGRN responses to 2 mM caffeine. (D) $Ih$ RNAi knockdown in sGRNs increased the sGRN responses to 50 mM sucrose. (E) Introduction of the $Ih$-RF cDNA in sGRNs, but not bGRNs, of $Ih^{f03355}$ restored the bGRN response to 2 mM caffeine. (F) For sucrose responses, the introduction of $Ih$-RF to bGRNs increased the spiking frequency. Letters indicate statistically distinct groups (a and b): Tukey's test, $p < 0.05$ (A), Dunn's, $p < 0.05$ (F). §: Welch's ANOVA, Games-Howell test, $p < 0.05$. #: Dunn's test, $p < 0.05$. Numbers in gray indicate the number of tested naïve bristles, which are from at least three individuals.

The online version of this article includes the following source data and figure supplement(s) for figure 1:

**Source data 1.** Spiking frequencies from the first 5-sec bin following the contact with indicated tastants, which are for box plots in the *Figure 1*.

**Figure supplement 1.** $Ih$ is required for spiking responses to various bitter chemical compounds.

**Figure supplement 1—source data 1.** The first 5-sec spiking frequencies in response to the indicated bitter compounds, which were used to draw the box plots.

**Figure supplement 2.** $Ih$ RNAi knockdown in adulthood reduces spiking frequencies in response to 2 mM caffeine but increases spiking frequencies to 50 mM sucrose.

**Figure supplement 2—source data 1.** The first 5-sec firing frequencies in response to the indicated tastants in *Figure 1—figure supplement 2B*.

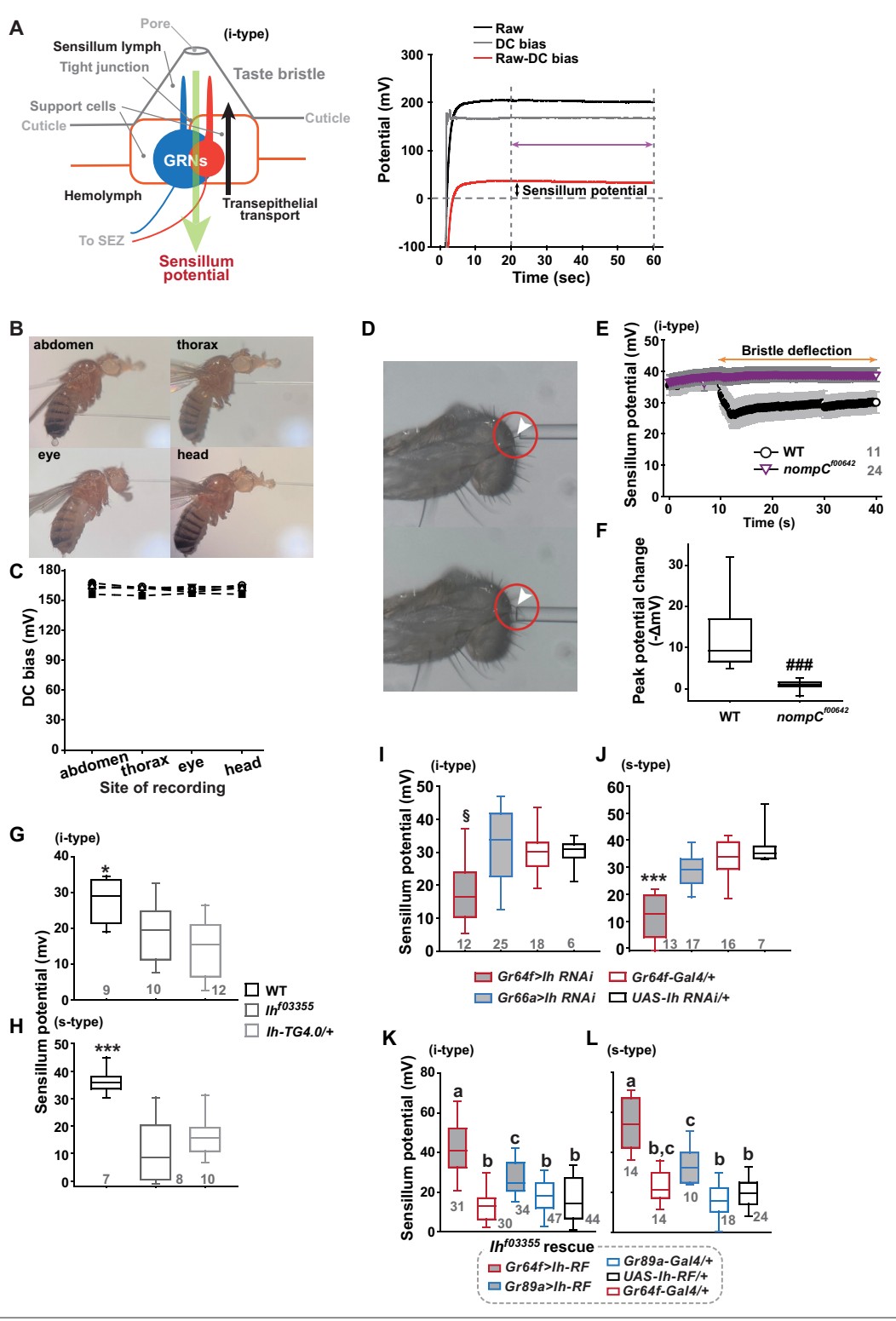

**Figure 2.** Sensillum potential (SP) is reduced in hyperpolarization-activated cyclic nucleotide-gated (HCN) channel-deficient animals. (**A**) Schematic diagram illustrating the sensillum potential in the taste bristle sensilla (Left). Black upward arrow indicates ion transport by pumps and transporters in support cells from the hemolymph to the sensillum lymph. These body fluids are physiologically separated by tight junctions between support cells. The resulting transcellular disparity of ions leads to a positive sensillum potential (greed downward arrow). Representative traces of potentials measured to evaluate SP (Right). Raw: the potential reading upon the contact

*Figure 2 continued on next page*

*Figure 2 continued*

of the recording electrode with the sensillum bristle tip (black). DC (direct current) bias: the potential reading upon impalement of the head by the recording electrode (gray). Red line indicates the difference between raw and DC bias, which represents the sensillum potential. The values resulting from the subtraction of the data between 20–60 s after the initial contact (time indicated by the purple double-headed arrow) were averaged to determine SP. (**B**) Photographs of impaled flies for DC bias determination at indicated sites. (**C**) DC bias values were obtained from indicated body parts. There is no statistical significance between the body sites (ANOVA Repeated Measures). (**D**) Photos before (top) and after (bottom) deflection of an i-type bristle. (**E**) Sensillum potential traces as a function of time from wild-type (WT) and *nompC$^{f00642}$*. Bristle bending started at 10 s, and the duration is marked by an orange double-headed arrow. (**F**) The peak SP changes of WT and *nompC$^{f00642}$* were compared. (**G, H**) SP was reduced in i- (**G**) and s-type (**H**) bristles of the indicated *Ih*-deficient mutants, relative to WT. (**I, J**) *Ih* RNAi in sweet-sensing GRNs (sGRNs) reduced SPs of the i- and s-type bristles. (**K, L**) The SP of *Ih$^{f03355}$* was restored by *Ih-RF* expression in gustatory receptor neurons (GRNs) (red for sGRNs, blue for bGRNs). ###: Dunn's, $p<0.001$. * and ***: Tukey's, $p<0.05$ and $p<0.001$, respectively. §: Welch's ANOVA, Games-Howell test, $p<0.05$. Letters indicate statistically distinct groups: Tukey's test, $p<0.05$. Numbers in gray indicate the number of naive bristles tested in at least three animals.

The online version of this article includes the following source data for figure 2:

**Source data 1.** Acquired potential values in indicated experiments in *Figure 2*.

Next, the sensillum potential was obtained by subtracting the DC bias from the raw potential value (*Figure 2A*). We also found that we could reduce the apparent SP by deflecting the bristle sensillum by ~45° (*Figure 2D–F*), activating the sensillum's mechanosensory neuron. When we performed the same experiment with *nompC$^{f00642}$*, a loss-of-function allele of *nompC* that encodes a mechanosensory TRPN channel (*Sánchez-Alcañiz et al., 2017*), this reduction in SP disappeared (*Figure 2D–F*).

Suggesting the role of *Ih* in SP regulation, *Ih$^{f03355}$* (~19 and ~10 mV for i-type and s-type sensilla, respectively) and *Ih-TG4.0/+* (~15 and ~16 mV for i-type and s-type sensilla, respectively) exhibited reduced mean SPs compared to WT in the i-type (*Figure 2G*) and s-type (*Figure 2H*) bristle sensilla (~28 mV and ~36 mV, respectively). We also examined whether the SP reduction could be attributed to the lack of *Ih* in sGRNs through GRN-specific *Ih* RNAi knockdown. This revealed that *Ih* is necessary in sGRNs for the sensilla to exhibit normal SP levels (*Figure 2I and J*). The SP reduction observed in both bristle types of *Ih$^{f03355}$* could be fully restored by expressing the *Ih-RF* cDNA in sGRNs (*Gr64f-Gal4* cells). Mean SPs were measured to be ~42 and ~54 mV in i-type and s-type bristles, respectively (*Figure 2K and L*). Interestingly, ectopic expression of the cDNA in bGRNs by *Gr89a-Gal4* also significantly rescued the SP defect of *Ih$^{f03355}$* to the level of mean SPs (~27 and ~33 mV in i-type and s-type bristles, respectively) comparable to those in WT. The greater extent of SP defect restoration in *Ih$^{f03355}$* by *Ih-RF* expressed in sGRNs than bGRNs indicates that *Ih-RF* is more effective at upholding SP in sGRNs than in bGRNs under our experimental conditions. Furthermore, the successful rescue by *Ih-RF* in bGRNs also shows that *Ih* can regulate SP in any GRN (*Figure 2K and L*).

## Inactivation of sGRNs raised both bGRN activity and SP, which was reversed by *Ih* deficiency

Since it is in sGRNs that HCN regulates the bGRN responsiveness to caffeine, we suspected that the activity of sGRNs may be closely associated with the maintenance of bGRN excitability. In line with this possibility, the *Gr64af* deletion mutant, which lacks the entire *Gr64* gene locus and is severely impaired in sucrose and glucose sensing (*Kim et al., 2018*; *Slone et al., 2007*; *Jiao et al., 2008*), showed increased bGRN responses to various bitters in labellar gustatory bristle sensilla compared to WT (*Figure 3A*). Furthermore, silencing sGRNs (*Gr5a-Gal4* cells) by expressing the inwardly rectifying potassium channel, Kir2.1 (*Baines et al., 2001*), phenocopied *Gr64af* in response to 2 mM caffeine stimulating the i-type bristles (*Figure 3B*). This increased responsiveness of bGRNs is unlikely due to positive feedback resulting from the sGRN inactivation through the neural circuitry in the brain, because the tetanus toxin light chain (TNT) expressed in sGRNs, which blocks chemical synaptic transmission (*Broadie et al., 1995*), failed to raise bGRN activity (*Figure 3C*). Strikingly, when we combined the sGRN-hindering genotypes (*Gr5a>Kir2.1* and *Gr64af*) with the *Ih* alleles *Ih$^{f03355}$* or *Ih-TG4.0*, we found that the sGRN inhibition-induced increase in bGRN activity in response to caffeine could be commonly relieved by the disruptions in the *Ih* gene (*Figure 3B and E*). This result suggests that HCN suppresses

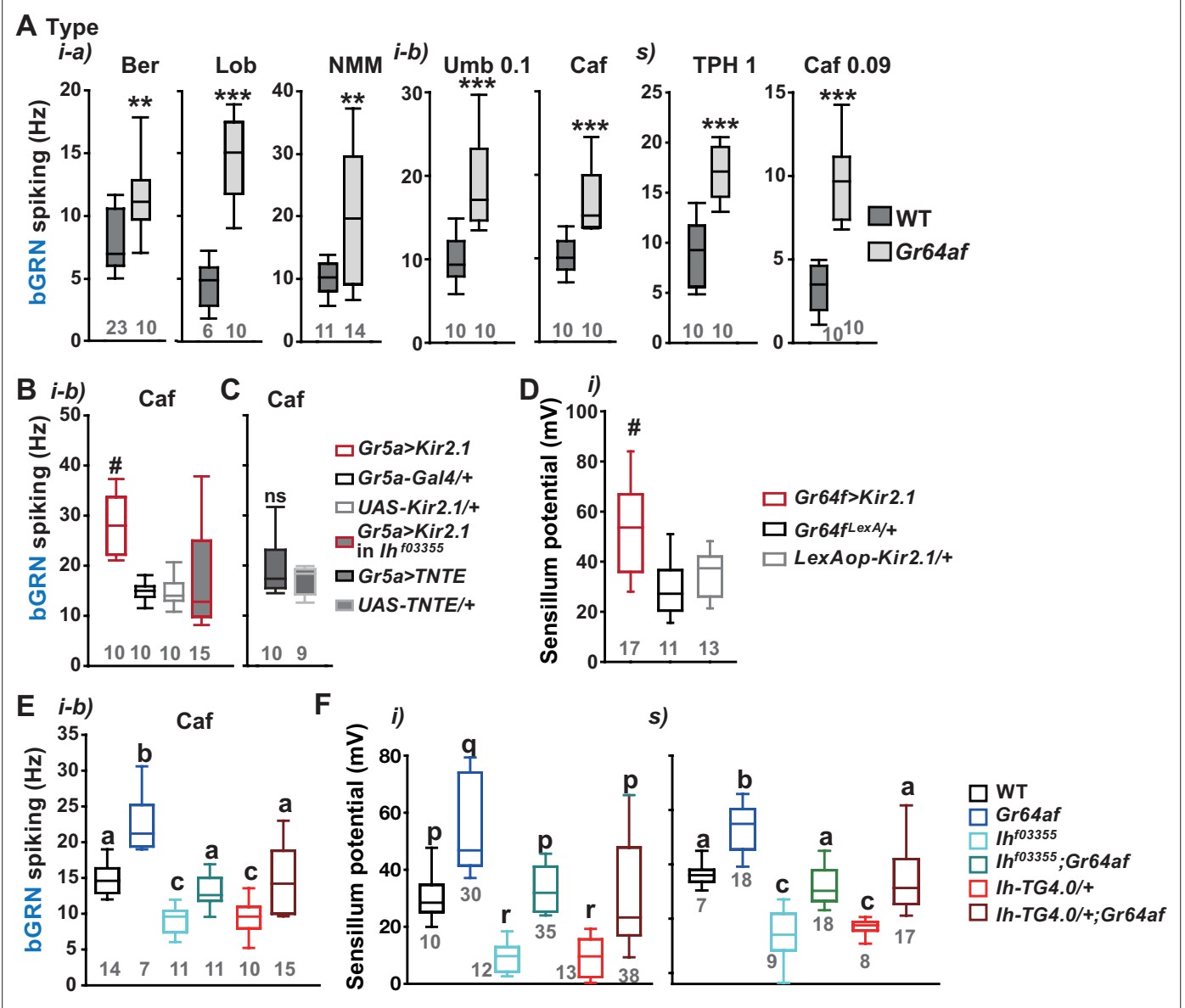

**Figure 3.** Inactivation of sweet-sensing GRNs (sGRNs) raises bitter-sensing gustatory receptor neurons (bGRN) activity and sensillum potential (SP), both of which are reversed by *Ih* deficiency. (**A**) The bGRN spiking was increased in response to the indicated bitters in *Gr64af* mutants impaired in sucrose and glucose sensing. Ber: 0.5, Lob: 0.5, NMM: 2, Caf: 2 (i-type), and 0.09 (s-type), Umb: 0.1, TPH: 1 mM. ** and ***: Student's t-test, $p<0.01$ and $p<0.001$, respectively. (**B, C**) Silencing by Kir2.1 (**B**), but not blocking chemical synaptic transmission (**C**), in sGRNs increased the spiking of bGRNs stimulated by 2 mM caffeine, which was reversed in *Ih^f03355^* (**B**). #: Dunn's, $p<0.05$. (**D**) Silencing sGRNs by Kir2.1 increased SP. #: Dunn's, $p<0.05$. (**E**) The increased bGRN spiking in *Gr64af* was restored to wild-type (WT) levels by *Ih* deficiencies. Letters indicate significantly different groups (Tukey's, $p<0.05$). Caffeine 2 mM was used (**B, C, E**). (**F**) Regardless of bristle type, SP was increased upon sGRN inactivation, which was reduced by *Ih* deficiencies. (p–r): Dunn's test, $p<0.05$. (a–c): Welch's ANOVA, Games-Howell test, $p<0.05$. Numbers in gray indicate the number of naïve tested bristles in at least three animals.

The online version of this article includes the following source data and figure supplement(s) for figure 3:

**Source data 1.** Spiking frequencies and sensillum potentials obtained in the experiments of *Figure 3*.

**Figure supplement 1.** Water gustatory receptor neurons (GRNs) rely on the sensillum potential (SP) guarded by hyperpolarization-activated cyclic nucleotide-gated (HCN) channel in the L-type bristles.

**Figure supplement 1—source data 1.** Water cell spiking frequencies and L-type bristle sensillum potential data.

sGRN activation, while HCN expressed in sGRNs is required for unimpaired bGRN activity (*Figure 1C and E*). Interestingly, Kir2.1-induced inactivation of sGRNs (*Gr64f-Gal4* cells) dramatically increased the mean SP of the i-type bristles to ~53 mV, compared to ~29 and ~35 mV of *Gal4* and *UAS* controls, respectively (*Figure 3D*), and the impairment of sucrose-sensing in the *Gr64af* mutants also resulted in increases of mean SPs (*Figure 3F*,~56 and~53 mV in the i- and s-bristles of *Gr64af*, compared to ~30 and ~36 mV of WT, respectively). Thus, inactivating sGRNs in two different ways increased SP in the i- and s-type gustatory bristles, similar to the effect on bGRN activity described earlier. Such repeated parallel shifts of bGRN activity and SP were again obtained in the combined genotypes between *Gr64af* and *Ih^{f03355}* or *Ih-TG4.0/+* (*Figure 3F*); the SP increased in *Gr64af* descended to WT levels when combined with *Ih^{f03355}* and *Ih-TG4.0/+*, similar to what occurred with bGRN activity in *Gr64af* (*Figure 3E*). These results suggest that *Ih* gene expression suppresses sGRNs, upholding both bGRN activity and SP, similar to the genetic alterations that reduce sGRN activity.

Water GRNs are co-housed with sGRNs in L-type bristles in the labellum, responding to hypo-osmolarity with the aid of *ppk28* and promoting water drinking (*Chen et al., 2010*; *Cameron et al., 2010*). We tested whether *Ih*-dependent SP regulation occurs in these bristles to maintain the sensitivity of water GRNs by using a low concentration of the electrolyte tricholine citrate (0.1 mM TCC). Interestingly, L-type bristles of *Ih^{f03355}* showed reduced spike frequencies in response to this hypo-osmolar electrolyte solution compared to WT (*Figure 3—figure supplement 1A*). This reduction was restored in the genetic rescue line. Additionally, SP in these bristles was increased in *Gr64af* but decreased in the two *Ih* alleles, and the combination of the *Gr64* and *Ih* mutations restored SP to the level of WT (*Figure 3—figure supplement 1B*), as observed with other sensillar bristles above. Finally, *Ih-RF* restored SP in *Ih^{f03355}* when expressed in sGRNs but not bGRNs, as expected from the absence of bGRNs in L types (*Figure 3—figure supplement 1C*). Thus, *Ih*-dependent SP regulation is universal in all bristle sensilla of the labellum and likely important for the function of GRNs neighboring sGRNs.

## HCN delimits excitability of HCN-expressing GRNs, and increases SP

By misexpressing *Ih-RF* in bGRNs of WT flies, we investigated how HCN physiologically controls HCN-expressing GRNs (*Figure 4A*). The genetic controls, *Gr89a-Gal4/+* and *UAS-Ih-RF/+*, exhibited mutually similar dose dependencies saturated at 2- and 10 mM caffeine, revealing the maximal caffeine responses at these concentrations. Interestingly, the ectopic expression reduced bGRN activity at these high caffeine concentrations (*Figure 4A*). The flattened dose dependence suggests that ectopically expressed HCN suppresses strong excitation of bGRNs. In contrast, sGRNs were upregulated by the misexpression of *Ih* in bGRNs with increased spiking in response to 10 and 50 mM sucrose (*Figure 4B*), implying that *Ih* increases the activity of the neighboring GRN by reducing that of *Ih*-expressing GRNs. On the other hand, the *Ih-RF*-overexpressing sGRNs in *Gr64f-Gal4* cells significantly decreased only the response 5 s after contacting 50 mM sucrose (*Figure 4C*, the second 5 s bin, *Figure 4—figure supplement 1*), probably because of native HCN preoccupying WT sGRNs. Although bGRNs were repressed by misexpressing *Ih-RF*, the mean SPs increased to ~40 and ~37 mV in the i- and s-type bristles, respectively, compared to controls with mean SPs of 22–25 mV (*Figure 4D*). These results from misexpression experiments corroborate the postulation that sGRNs are suppressed by expressing HCN. To confirm that sGRNs are suppressed by native HCN, the impact of GRN-specific *Ih* RNAi knockdown on sGRNs was quantitatively evaluated (*Figure 4E*). *Ih* RNAi in sGRNs (*Gr64f-Gal4* cells) led to increased mean spiking frequencies by ~10 Hz in response to 1-, 5-, and 10 mM as well as 50 mM sucrose compared to *Ih* RNAi in bGRNs (*Gr66a-Gal4* cells) and genetic controls, highlighting the extent to which HCN natively expressed in sGRNs suppresses sGRN excitability. In contrast, SP, necessary for GRN sensitization, was observed above to be reduced by *Ih* RNAi in sGRNs but not bGRNs (*Figure 2I and J*). Thus, these data suggest that HCN innately reduces the spiking frequencies of sGRNs even at relatively low sucrose concentrations, 1, and 5 mM. This is similar to the suppressive effect of *Ih-RF* misexpressed in bGRNs at relatively high caffeine concentrations, but differs in that the misexpression did not alter bGRN activity in response to low caffeine concentrations, 0.02 and 0.2 mM (*Figure 4A*), implying a complex cell-specific regulation of GRN excitability.

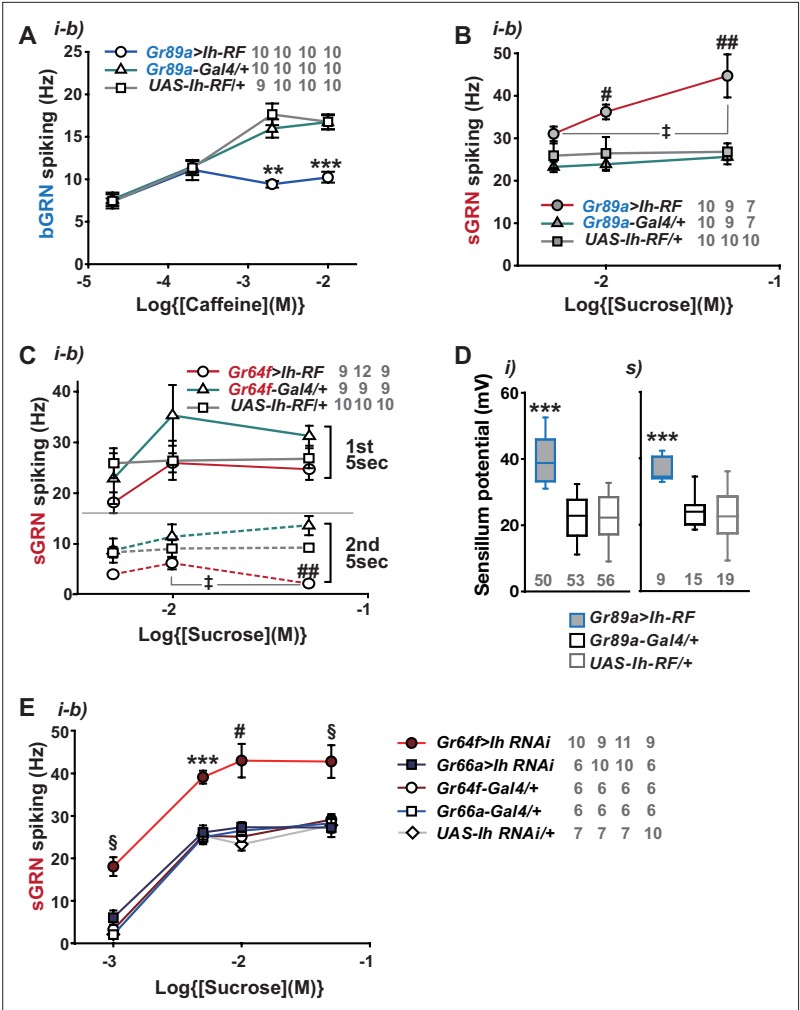

**Figure 4.** Hyperpolarization-activated cyclic nucleotide-gated (HCN) channel suppresses HCN-expressing gustatory receptor neurons (GRNs) and increases sensillum potential (SP). (**A**) HCN misexpressed in bitter-sensing gustatory receptor neurons (bGRNs) flattened the dose dependence to caffeine. (**B**) HCN ectopically expressed in bGRNs elevates sweet-sensing GRN (sGRN) responses to sucrose. (**C**) Overexpression of HCN in sGRNs reduced the sGRN responses to sucrose 5 s after the initial contact. (**D**) *Ih* misexpression in bGRNs increased SP in i- and s-type bristles, which correlates with laterally increased sGRN activity (**B**). (**E**) *Ih* RNAi knockdown in sGRNs (*Gr64f-Gal4* cells) dramatically elevates spiking frequencies in response to 1-, 5-, 10-, and 50 mM sucrose. *, **, and ***: Tukey's, $p<0.05$, $p<0.01$, and $p<0.001$, respectively (**A, D, E**). # and ##: Dunn's, $p<0.05$ and $p<0.01$ between genotypes, respectively (**B, C, E**). ‡: Dunn's, $p<0.05$ between responses to different sucrose concentrations (**B, C**). §: Welch's ANOVA, Games-Howell test, $p<0.05$ (**E**). The numbers in gray indicate the number of tested naïve bristles in at least three animals.

The online version of this article includes the following source data and figure supplement(s) for figure 4:

**Source data 1.** Spiking frequencies and sensillum potential data from *Figure 4*.

**Figure supplement 1.** Overexpression of *Ih-RF* in WT sGRNs suppresses their spiking responses to 50 mM sucrose in a delayed manner.

**Figure supplement 1—source data 1.** Post-stimulus spiking frequenceis in 1-sec bins.

## Sweetness in the food leads to a reduction of SP, bGRN activity, and bitter avoidance in *Ih*-deficient animals

Typically, we performed extracellular recordings on flies 4–5 days after eclosion, during which they were kept in a vial with fresh regular cornmeal food containing ~400 mM D-glucose. The presence of sweetness in the food would impose strong and frequent stimulation of sGRNs

for an extended period, potentially requiring the delimitation of sGRN excitability for the homeostatic maintenance of gustatory functions. To investigate this possibility, we fed WT and $Ih^{f03355}$ flies overnight with either non-sweet sorbitol alone (200 mM) or a sweet mixture of sorbitol (200 mM)+sucrose (100 mM). Although sorbitol is not sweet, it is a digestible sugar that provides *Drosophila* with calories (*Fujita and Tanimura, 2011*). We found that the sweet sucrose medium significantly reduced caffeine-induced bGRN responses in both genotypes compared to the sorbitol-only medium, but $Ih^{f03355}$ bGRN spike frequencies were decreased to a level significantly lower than WT (*Figure 5A*), as seen above with the cornmeal food (*Figures 1A, C , and 3E*). This suggests that the reduced bGRN activity in the mutants may result from prolonged sGRN excitation. The SP reduction was similarly induced by 1 hr incubation with the sweet sucrose medium in both WT and $Ih^{f03355}$. However, the *Ih* mutant showed a more severe depletion of SPs compared to WT after 4 hr of sweet exposure (*Figure 5B*) as observed with the cornmeal food (*Figures 2 and 3F*). Even on the sorbitol food, the SP in $Ih^{f03355}$ was significantly decreased compared to WT. This may be attributed to the loss of HCN, which is known to stabilize the resting membrane potential (*Shah, 2014*). Following overnight sweet exposure, SPs of WT and $Ih^{f03355}$ were recovered to similar levels after 1 hr incubation with sorbitol-only food. However, it was after 4 hr on the sorbitol food that the two lines exhibited SP levels similar to those achieved by overnight incubation with sorbitol-only food (*Figure 5B*). These results indicate that SP depletion by sweetness is a slow process, and that the dysregulated reduction and recovery of SPs in $Ih^{f03355}$ manifest only after long-term conditioning with and without sweetness, respectively.

To assess the behavioral implications of HCN-assisted preservation of SP and bGRN activity, flies were exposed long-term to sweetness on a regular sweet cornmeal diet (sweet exposure-positive), and then subjected to a CAFE with an 8 hr choice between water and 4 mM caffeine solution. Note that sucrose was not used in CAFE, because the presence of sweet stimuli was shown to suppress bGRNs (*Lee et al., 2023*). Indicative of reduced bitter sensitivity, $Ih^{f03355}$ flies showed dramatically decreased caffeine avoidance, relative to WT (*Figure 5C*). In contrast, when flies were removed from the cornmeal food for 20 hr, both WT and $Ih^{f03355}$ showed similarly robust bitter avoidance. The defect observed in the *Ih* mutant on the sweet cornmeal diet could be rescued by reintroducing a genomic fragment covering the *Ih* locus ({*Ih*}). These results were recapitulated with other bitters, lobeline, and theophylline (*Figure 5—figure supplement 1*). To examine whether caffeine avoidance requires *Ih* expression in sGRNs, CAFE was performed with GRN-specific RNAi knockdown of *Ih*. For the RNAi experiments, flies were kept overnight on either the non-sweet diet with sorbitol (200 mM) or the sweet diet with additional sucrose (100 mM). *Ih* knockdown in sGRNs, but not bGRNs, led to a deficit in the avoidance only when the flies had been on the sweet diet, indicating that HCN expression in sGRNs is necessary for robust caffeine avoidance in a sweet environment (*Figure 5D*). Therefore, the sweetness of the diet can compromise the function of bGRNs co-housed with sGRNs in the same sensilla, which is mitigated by HCN expression in sGRNs. Such a role of HCN is essential for bitter avoidance of flies, considering their likely prolonged exposure to sweetness in their natural habitat of overripe fruit (*Figure 5E*).

## Discussion

Our results provide multiple lines of evidence that HCN suppresses HCN-expressing GRNs, thereby sustaining the activity of neighboring GRNs within the same sensilla (*Figure 5F*). We propose that this modulation occurs by restricting SP consumption through HCN-dependent neuronal suppression rather than via chemical and electrical synaptic transmission. The lack of increased bGRN activity with TNT expression in sGRNs, coupled with the increase observed with Kir2.1 expression (*Figure 3B and C*), indicates minimal involvement of synaptic vesicle-dependent transmission. The possibility of a neuropeptide-dependent mechanism is unlikely, given our ectopic gain-of-function studies (*Figure 4*). To explain the misexpression results with neuropeptide pathways, both s- and bGRNs must be equipped with the same set of a neuropeptide/receptor system, which is incompatible with the inverse relationship between the two GRNs in excitability observed in *Figure 1C and D*, *Figure 1—figure supplement 2B*, and *Figure 3*. Furthermore, this inverse relationship argues against electrical synapses through gap

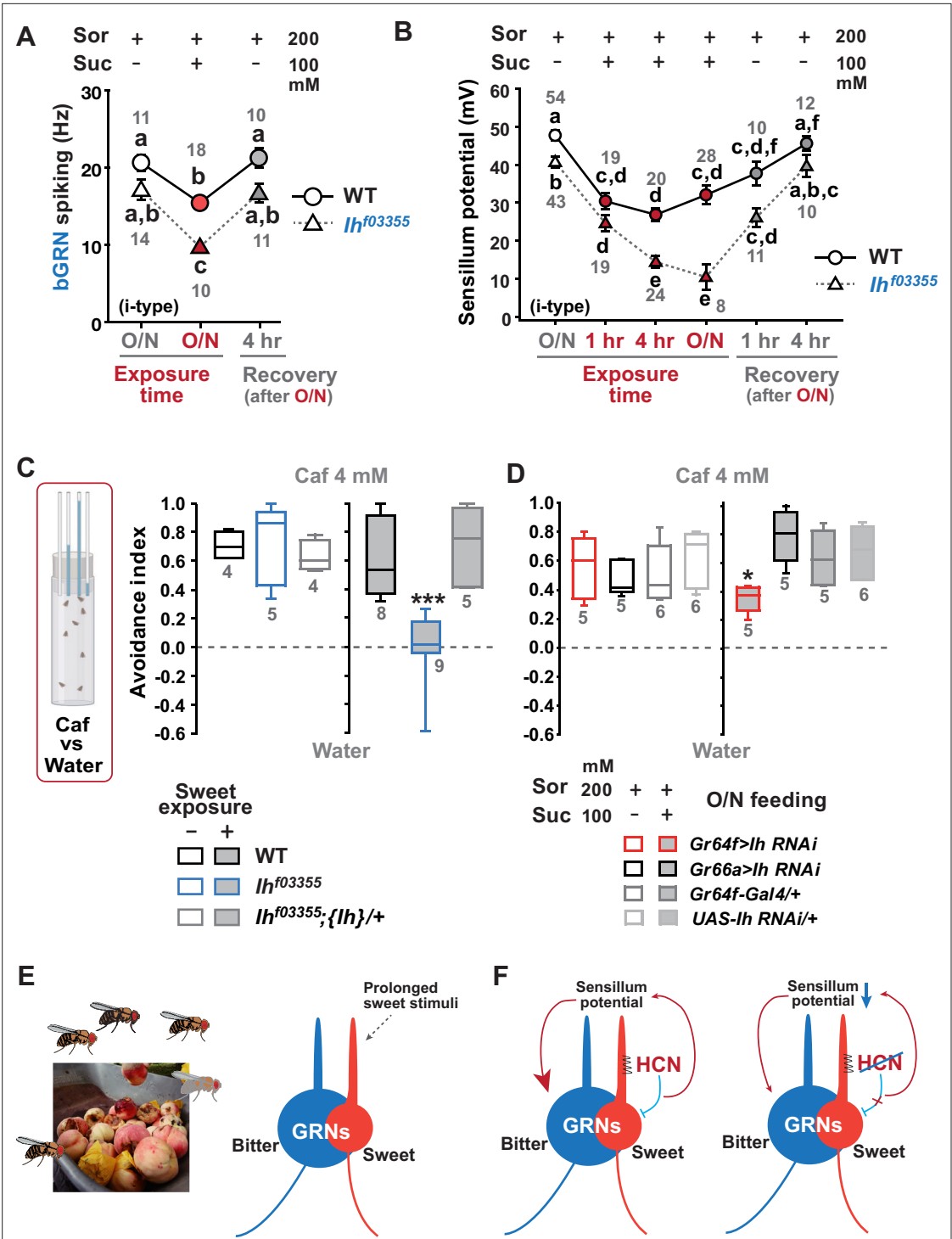

**Figure 5.** Sweetness in the diet decreases sensillum potential (SP), bitter-sensing gustatory receptor neuron (bGRN) activity, and bitter avoidance. (**A**) Sweetness in the media reduced the 2 mM caffeine-evoked bGRN spiking, which was fully recovered in 4 hr incubation with sorbitol only food. *Ih^f03355* was affected by the type of the media more severely than wild-type (WT). O/N: overnight incubation with sorbitol only (gray) or sucrose food (red). (**B**) The SP of *Ih^f03355* bristle sensilla showed dysregulated reduction after 4 hr and overnight incubation on sweet media. These reductions started to be recovered in 1 hr feeding and were nearly fully recovered in 4 hr feeding on the indicated sorbitol only food. (**C**) Caffeine (Caf) avoidance was assessed with capillary feeder assay (CAFE). *Ih* is required for robust caffeine avoidance for flies maintained on sweet cornmeal food (sweet exposure +: filled boxes). *Ih^f03355* flies avoided 4 mM caffeine like WT flies when separated from sweet food for 20 hr (blank boxes). (**D**) *Ih* RNAi knockdown in sGRNs (*Gr64f-Gal4*) but not bGRNs (*Gr66a-Gal4*) led to relatively poor avoidance to caffeine after feeding on the sweet diet with sucrose. Suc: sucrose, and Sor: sorbitol. Letters indicate statistically distinct groups: a-f, Dunn's, p<0.05 (**A, B**). * and ***: Tukey's, p<0.05 and<0.001, respectively. (**E**) Illustration

*Figure 5 continued on next page*

*Figure 5 continued*

depicting the flies' sweet feeding niche in overripe fruit (Left), leading to prolonged exposure of sGRNs to the sweetness (Right). (**F**) A schematic model of gustatory homeostasis in *Drosophila* bristle sensilla. Despite the prolonged sweetness in the environment robustly and frequently stimulating sweet-sensing GRNs (sGRNs), the sGRN activity is moderated by hyperpolarization-activated cyclic nucleotide-gated (HCN) channel to preserve the sensillum potential, which is required for normal bGRN responsiveness (Left). When HCN in sGRNs is incapacitated, sGRNs can become overly excited by sweetness of overripe fruit and deplete the sensillum potential, resulting in decreased bGRN activity and bitter avoidance (Right).

The online version of this article includes the following source data and figure supplement(s) for figure 5:

**Source data 1.** Electrophysiology data and avoidance indices.

**Figure supplement 1.** Feeding avoidance to lobeline and theophylline is reduced in *Ih^f03355* following prior exposure to sweetness.

**Figure supplement 1—source data 1.** Avoidance indices obtained with indicated bitters.

junctions, which typically synchronize the excitability of pre-and postsynaptic neurons. Therefore, our findings propose an unconventional mechanism of neuronal interaction.

HCNs are encoded by four different genes in mammals (***Shah, 2014***; ***Biel et al., 2009***), and are known to be present in mammalian sensory receptor cells. In cochlear hair cells, HCN1 and HCN2 were reported to form a complex with a stereociliary tip-link protein (***Ramakrishnan et al., 2012***), while in vestibular hair cells, HCN1 is essential for normal balance (***Horwitz et al., 2011***). HCN1 was also immunostained in cone and rod photoreceptors, as well as retinal bipolar, amacrine, and ganglion neurons, with deletion of the encoding gene resulting in prolonged light responses (***Knop et al., 2008***). A subset of mouse taste cells was labeled for HCN1 and HCN4 transcripts and proteins (***Stevens et al., 2001***), similar to our observation of selective HCN expression in *Drosophila* GRNs. HCN2 is expressed in small nociceptive neurons that mediate diabetic pain (***Tsantoulas et al., 2017***). However, the precise roles of HCNs in regulating these respective sensory physiologies remain to be elucidated.

HCN is well-known for its 'funny' electrophysiological characteristics, stabilizing the membrane potential (***Shah, 2014***; ***Biel et al., 2009***). As a population of HCN channels remains open at the resting membrane potential, HCN serves to suppress neuronal excitation in two ways. First, it increases the inward current required to depolarize the membrane and trigger action potentials, owing to the low membrane input resistance resulting from the HCN-dependent passive conductance. Second, the closing of HCN induced by membrane depolarization counteracts the depolarization, since the reduction of the standing cation influx through HCN is hyperpolarizing. Conversely, HCNs also allow neurons to resist membrane hyperpolarization because the hyperpolarization activates HCNs to conduct depolarizing inward currents. Consequently, HCN channels effectively dampen fluctuations in membrane potential, whether they lead to depolarization or hyperpolarization. Our findings in this study align with the former property of HCNs, as *Drosophila* HCN is essential for moderating sGRN excitation to preserve SP and bGRN activity when flies inhabit in sweet environments. On the other hand, our previous study showed that HCN-dependent resilience to hyperpolarizing inhibition of sGRNs lateralizes gustatory ephaptic inhibition to dynamically repress bGRNs, when exposed to strong sweetness together with bitterness (***Lee et al., 2023***). Thus, depending on the given feeding contexts, the electrophysiological properties of HCN in sGRNs lead to playing dual roles with opposing effects in regulating bGRNs. The stabilization of membrane potential by HCNs was reported to decrease the spontaneous activity of neurons, as evidenced by miniature postsynaptic currents suppressed by presynaptic HCNs (***Cai et al., 2022***). In this regard, the lower SP observed in the *Ih^f03355* labellar bristles than that of WT, even on the nonsweet sorbitol food (***Figure 5B***), may be attributed to the more facile fluctuations in resting membrane potential which could regulate the consumption of SP (further discussion below).

Cell-specific knockdown of *Ih* in sGRNs led to increased sGRN responses to 50 mM sucrose (***Figure 1D***), although disruptions of the *Ih* locus did not (***Figure 1B***). This inconsistency may stem from differences between alleles and the RNAi knockdown in residual *Ih* expression or in *Ih*-deficient sites. The lack of *Ih* in sGRNs can induce two different effects in neuronal excitation: (1) easier depolarization of sGRNs due to the loss of standing HCN currents at rest as suggested in ***Figure 4E*** and (2) a decrease of receptor-mediated inward currents, expected due to SP reductions (***Figure 2G–L***). Assuming that some level of HCN expression may persist in RNAi knockdowns compared to mutants, these opposing effects on sGRN excitability may

largely offset each other in response to 50 mM sucrose in the *Ih* mutants, but not in the knock-downed flies. The ectopic introduction of *Ih-RF* into bGRNs of *Ih^f03355^* significantly increased the mean SP compared to the control genotypes (*Figure 2K and L*), leaving sGRNs devoid of functional *Ih*. This genotype allows the examination of sGRNs lacking *Ih*, with SP unimpaired, which is supposed to reflect the net effect of *Ih* on sGRN excitability excluding the influence from reduced SP. Interestingly, the ectopic rescue resulted in elevated firing responses to 50 mM sucrose compared to the cDNA rescue in sGRNs (*Figure 1F*), a proper control with *Ih* expression and SP both unimpaired. On the other hand, the differing sites of *Ih* deficiency might create the inconsistency. The protein trap reporter *Ih-TG4.0-Gal4* previously showed widespread expression of HCN in the labellum, including non-neuronal cells, implying the possibility of unknown bGRN-regulating HCN-dependent mechanisms, potentially harbored in nonneuronal cells. Overall, our cell-specific loss-of-function and gain-of-function studies advocate that HCN suppresses HCN-expressing GRNs, which thereby increases SP to promote the activity of the neighboring GRNs.

Only the dendrites of GRNs face the sensillar lymph, separated from the hemolymph by tight junctions between support cells (*Shanbhag et al., 2001*). The inward current through the ion channels that respond to sensory reception in the dendrites is thought to be a major sink for SP (*Tuthill and Wilson, 2016*; *Syed and Leal, 2008*), consistent with the incremented SP in the *Gr64af* mutant lacking the sucrose-sensing molecular receptor (*Lee et al., 2023*; *Kim et al., 2018*). Based on these points, it was somewhat unexpected that the membrane potential regulator HCN preserved SP, yet implying that the sensory signaling in the dendrite is likely under voltage-dependent control. In line with HCN, shifting the membrane potential toward the K$^+$ equilibrium by overexpressing Kir2.1 in sGRNs upregulated bGRN activity and SP (*Figure 3*), corroborating that the membrane potential in sGRNs is a regulator of the sensory signaling cascade in the dendrites. Note that the sensillum lymph contains high [K$^+$] (*Tuthill and Wilson, 2016*; *Sollai et al., 2008*), which would not allow strong inactivation of sGRNs and SP increases if Kir2.1 operates mostly in the dendrites. The increases in SP, coinciding with the apparent silencing of sGRNs by Kir2.1 (*Lee et al., 2023*), propose that lowering the membrane potential in the soma and the axon suppresses the consumption of SP probably by inhibiting the gustatory signaling-associated inward currents in the dendrite. Para, the *Drosophila* voltage-gated sodium channel, was reported to be localized in the dendrites of mechanosensitive receptor neurons in *Drosophila* chordotonal organs (*Ravenscroft et al., 2023*). Similarly, *Drosophila* voltage-gated calcium channels have been studied in dendrites (*Kanamori et al., 2013*; *Ryglewski et al., 2012*; *Kadas et al., 2017*), implying that membrane potential may be an important contributor to the sensory signaling in dendrites.

There are ~14,500 hair cells in the human cochlea at birth (*Ashmore, 2008*). These hair cells share the endolymph in the scala media (cochlear duct), representing a case of TEP shared by a large group of sensory receptor cells. Since HCNs were found to be unnecessary for mechanotransduction itself in the inner ear (*Horwitz et al., 2010*), they may play a regulatory role in fine-tuning the balance between the endocochlear potential maintenance and mechanotransduction sensitivity for hearing, as in the *Drosophila* gustatory system. Multiple mechanosensory neurons are found to be co-housed also in *Drosophila* mechanosensory organs such as hair plates and chordotonal organs (*Tuthill and Wilson, 2016*). Given that each mechanosensory neuron is specifically tuned to detect different mechanical stimuli such as the angle, velocity, and acceleration of joint movement (*Mamiya et al., 2018*), some elements of these movements may occur more frequently and persistently than others in a specific ecological niche. Such biased stimulation would require HCN-dependent moderation to preserve the sensitivity of other mechanoreceptors sharing the sensillar lymph. We showed that ectopic expression of *Ih* in bGRNs also upheld SP and the activity of the neighboring sGRNs, underscoring the independent capability of HCN in SP preservation. Despite such an option available, the preference for sGRNs over bGRNs in HCN-mediated taste homeostasis implies that *Drosophila melanogaster* may have ecologically adapted to the high sweetness (*Durkin et al., 2021*) prevalent in their feeding niche, such as overripe or fermented fruits (*Wang et al., 2022*). It would be interesting to investigate whether and how respective niches of various insect species differentiate the HCN expression pattern in sensory receptor neurons for ecological adaptation.

In this report, we introduce a peripheral coding design for feeding decisions that relies on HCN. HCN operating in sGRNs allows uninterrupted bitter avoidance, even when flies reside in sweet environments. This is achieved in parallel with an ephaptic mechanism of taste interaction by the same HCN in sGRNs, whereby bitter aversion can be dynamically attenuated in the simultaneous presence of sweetness (*Lee et al., 2023*). Further studies are warranted to uncover similar principles of HCN-dependent adaptation in other sensory contexts. It would also be interesting to explore whether the role of HCN in the sensory adaptation consistently correlates with lateralized ephaptic inhibition between sensory receptors, given that sensory cells expressing HCN can resist both depolarization and hyperpolarization of the membrane.

# Materials and methods

**Key resources table**

| Reagent type (species) or resource | Designation | Source or reference | Identifiers | Additional information |
|---|---|---|---|---|
| Genetic reagent (*Drosophila melanogaster*) | Cantonized w1118 | NA | NA | NA |
| Genetic reagent (*D. melanogaster*) | Gr64af | Dr. Moon at Yonsei U. | NA | NA |
| Genetic reagent (*D. melanogaster*) | Ihf03355 | Bloomington *Drosophila* Stock Center | BDSC: 85660; Flybase: FBti0051182 | NA |
| Genetic reagent (*D. melanogaster*) | Mi{Trojan-GAL4.0}IhMI03196-TG4.0 (Ih-TG4.0) | Bloomington *Drosophila* Stock Center | BDSC: 76162; Flybase: FBti0187533 | NA |
| Genetic reagent (*D. melanogaster*) | Duplicate of Dp(2;3)GV-CH321-22I11 | Bloomington *Drosophila* Stock Center | BDSC: 89744; Flybase: FBab0048672 | NA |
| Genetic reagent (*D. melanogaster*) | Gr5a-Gal4 | Dr. Scott at UC Berkeley | NA | NA |
| Genetic reagent (*D. melanogaster*) | Gr64fLexA | Dr. Amrein at TAMU | NA | NA |
| Genetic reagent (*D. melanogaster*) | Gr64f-Gal4 | Dr. Amrein at TAMU | NA | NA |
| Genetic reagent (*D. melanogaster*) | Gr89a-Gal4 | Dr. Carlson at Yale | NA | NA |
| Genetic reagent (*D. melanogaster*) | Gr66a-Gal4 | Dr. Amrein at TAMU | NA | NA |
| Genetic reagent (*D. melanogaster*) | UAS-Kir2.1 | Bloomington *Drosophila* Stock Center | BDSC: 6595 | NA |
| Genetic reagent (*D. melanogaster*) | LexAop-Kir2.1 | Dr. Dickson at Janellia | NA | NA |
| Genetic reagent (*D. melanogaster*) | UAS-TNTE | Bloomington *Drosophila* Stock Center | BDSC: 28837; Flybase: FBst0028837 | NA |
| Genetic reagent (*D. melanogaster*) | tub-Gal80ts | Bloomington *Drosophila* Stock Center | NA | NA |
| Genetic reagent (*D. melanogaster*) | UAS-Ih-RF | This study or doi: 10.1101/2023.08.04.551918 | Flybase: FBtr0290109 | NA |
| Genetic reagent (*D. melanogaster*) | UAS-Ih RNAi | Bloomington *Drosophila* Stock Center | BDSC: 58089; Flybase: FBst0058089 | NA |
| Genetic reagent (*D. melanogaster*) | nompCf00642 | Korea *Drosophila* Resource Center | KDRC: K3137; Flybase: FBt0041920 | NA |
| Chemical compound, drug | Tricholine citrate | Sigma-Aldrich | Cat. #T0252 | |
| Chemical compound, drug | Caffeine | Sigma-Aldrich | Cat. #C0750 | |
| Chemical compound, drug | Berberine chloride form | Sigma-Aldrich | Cat. #B3251 | |

*Continued on next page*

*Continued*

| Reagent type (species) or resource | Designation | Source or reference | Identifiers | Additional information |
|---|---|---|---|---|
| Chemical compound, drug | Lobeline hydrochloride | Sigma-Aldrich | Cat. #141879 | |
| Chemical compound, drug | Umbelliferone | Sigma-Aldrich | Cat. #H24003 | |
| Chemical compound, drug | Theophylline anhydrous | Sigma-Aldrich | Cat. #T1633 | |
| Chemical compound, drug | Sucrose | Georgia Chem | Cat. #57-50-1 | |
| Chemical compound, drug | D-Sorbitol | Sigma-Aldrich | Cat. #S1876 | |
| Chemical compound, drug | N-methyl maleimide | Sigma-Aldrich | Cat. #389412 | |
| Software, algorithm | LabChart 8 | AD Instrument | https://www.adinstruments.com | NA |
| Software, algorithm | SigmaPlot 14.0 | Systat Software Inc | https://systatsoftware.com/ | NA |

## Fly strains

The $w^{1118}$ line in a Canton-S background was used as wild-type. *Gr64f-Gal4* was provided by Dr. Hubert Amrein, and *Gr5a-Gal4* by Dr. Kristin Scott. *Gr64af* is a gift from Dr. Seok Jun Moon. *UAS-Ih RNAi* (#58089), a duplicate of the *Ih* locus (denoted as {Ih} in the main text, #89744), *Ih^{f03355}* (#85660), and *Ih-TG4.0* (#76162) were acquired from Bloomington *Drosophila* Stock Center (#stock number). The *UAS-Ih-RF* line was previously generated by the Korea *Drosophila* Resource Center (http://kdrc.kr) by site-specific recombination into attP49b (3 R), for which we cloned *Ih* cDNA through reverse transcription (*Lee et al., 2023*).

## Extracellular recordings

In vivo extracellular recordings were performed by the tip-dip method as detailed previously (*Hodgson et al., 1955*; *Du et al., 2016*). Each of the i-a, i-b, and s-b type sensillum of 3–5 day-old flies were identified from the sensillum map described elsewhere (*Weiss et al., 2011*). The reference electrode was filled with HL3.1 solution (*Feng et al., 2004*). The recording electrode contained tastants solubilized in the electrolyte 2 (i-type) or 30 (L- and s-type) mM tricholine citrate (TCC). The concentrations of bitter chemicals were indicated in the corresponding figure legend. The spiking frequency (Hz) was calculated from the number of spikes in the first 5 s or the second 5 s as indicated, and compared between genotypes or experimental conditions. The signals picked up by the electrodes were amplified by the preamplifier Tasteprobe (Syntech) and digitized at a rate of 20 kb/s by PowerLab with Labchart software (ADInstruments). The number of experiments indicated in the figures are the number of naïve bristles tested. The naïve bristles were from at least three different animals.

## Sensillum potential recordings

Media with or without sweetness were prepared as follows; the sorbitol medium consisted of 0.5% agarose and 200 mM sorbitol, while the sweet medium contained 0.5% agarose, 200 mM sorbitol, and 100 mM sucrose. Flies were kept overnight on these media before the experiment. For SP recordings, the recording electrode contained 2 mM TCC as the electrolyte, and Tasteprobe was set to record in 'pass-through' mode with DC (in the High-Pass filter window) and 100 ms zeroing time settings. Amplified signals were digitized at a rate of 100 Hz using PowerLab/Labchart. First, differential potentials were measured between a recording electrode on a taste sensillum and a reference electrode inserted into the labellum as performed for the extracellular bristle sensillum recordings. DC bias (*Marion-Poll and van der Pers, 1996*) was measured by impaling the recording electrode into the thorax of the same animals used for SP measurements. DC bias was subsequently subtracted from the initial read-outs of the differential potential to evaluate SP (*Figure 2A*). The resulting SPs were averaged during a 40 s long recording 20 s after initial contact, which was subsequently used for further analyses.

## Bitter avoidance assay

Twenty flies, aged 3–5 days and consisting of 10 males and 10 females, were used to assess bitter avoidance using capillary feeder assay (CAFE). To test the bitter sensitivity of each genotype of interest

in feeding behavior, flies were kept on regular cornmeal food or starved on nonsweet water-soaked Kimwipes overnight, and then given a choice between water and 4 mM caffeine for 8 hr. For RNAi experiments, 200 mM sorbitol is used in nonsweet food and sweet food, the latter of which included 100 mM sucrose in addition. Avoidance indices were obtained as the net volume fraction of water consumption subtracted by the volume fraction of caffeine ingestion.

## Statistics

Statistical calculation was performed using Sigmaplot 14.5 (Systat Software). The sample sizes and the statistical tests are indicated in each figure or in the legend. Normal distribution and heteroskedasticity were assessed using Shapiro-Wilk and Brown-Forsythe tests, respectively, before parametric tests. When these tests were failed, non-parametric tests were performed. However, for some cases, heteroskedasticity with normality led us to perform Welch's t-test (Sigmaplot 14) or Welch's ANOVA. The latter was followed by the Games-Howell test as a parametric analysis using the Excel spreadsheet available at https://www.biostathandbook.com/welchanova.xls. No outlier was excluded for statistical analyses.

## Acknowledgements

We would like to thank Drs Paul Garrity at Brandeis University and Kyuhyung Kim at DGIST for helpful comments, Drs. Amrein, H, Scott, K, Moon SJ, and KDRC/BDRC stock/resource centers for sharing fly lines as indicated in Materials and methods, and, for funding, National Research Foundation of Korea (NRF-2021R1A2B5B01002702, 2022M3E5E8017946 to KJK) and Korea Brain Research Institute (23-BR-01–02, 22-BR-03–06 to KJK), funded by Ministry of Science and ICT. This work is also indebted to the support from Brain Research Core Facilities at KBRI.

## Additional information

### Funding

| Funder | Grant reference number | Author |
|---|---|---|
| National Research Foundation of Korea | 2021R1A2B5B01002702 | KyeongJin Kang |
| National Research Foundation of Korea | 2022M3E5E8017946 | KyeongJin Kang |
| Korea Brain Research Institute | 23-BR-01-02 | KyeongJin Kang |
| Korea Brain Research Institute | 22-BR-03-06 | KyeongJin Kang |

The funders had no role in study design, data collection and interpretation, or the decision to submit the work for publication.

### Author contributions

MinHyuk Lee, Conceptualization, Investigation, Visualization, Methodology, Writing - original draft; Se Hoon Park, Investigation, Visualization, Methodology, Writing - review and editing; Kyeung Min Joo, Supervision, Investigation, Project administration, Writing - review and editing; Jae Young Kwon, Supervision, Methodology, Project administration, Writing - review and editing; Kyung-Hoon Lee, Investigation, Visualization, Project administration; KyeongJin Kang, Conceptualization, Supervision, Funding acquisition, Investigation, Visualization, Methodology, Writing - original draft

### Author ORCIDs
MinHyuk Lee ⬤ https://orcid.org/0009-0007-6101-5053
KyeongJin Kang ⬤ https://orcid.org/0000-0003-0446-469X

Reviewer #1 (Public Review): https://doi.org/10.7554/eLife.96602.3.sa1

Reviewer #2 (Public Review): https://doi.org/10.7554/eLife.96602.3.sa2
Reviewer #3 (Public Review): https://doi.org/10.7554/eLife.96602.3.sa3
Author response https://doi.org/10.7554/eLife.96602.3.sa4

## Additional files

### Supplementary files
• MDAR checklist

### Data availability
All data generated during this study are accompanied as source data files.

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
