## [Editor Report · eLife assessment]

This study provides **important** new insight into how non-synaptic interactions affect the activity of adjacent gustatory neurons housed within the same sensillum. The conclusions are supported by **convincing** electrophysiological, behavioral, and genetic data. This work will be of interest to neuroscientists studying chemosensory processing or regulation of neuronal excitability.

---

## [Referee Report · Reviewer #1 (Public Review)]

Summary:

This study identifies new types of interactions between *Drosophila* gustatory receptor neurons (GRNs) and shows that these interactions influence sensory responses and behavior. The authors find that HCN, a hyperpolarization-activated cation channel, suppresses the activity of GRNs in which it is expressed, preventing those GRNs from depleting the sensillum potential, and thereby promotes the activity of neighboring GRNs in the same sensilla. HCN is expressed in sugar GRNs, so HCN dampens excitation of sugar GRNs and promotes excitation of bitter GRNs. Impairing HCN expression in sugar GRNs depletes the sensillum potential and decreases bitter responses, especially when flies are fed on a sugar-rich diet, and this leads to decreased bitter aversion in a feeding assay. The authors' conclusions are supported by genetic manipulations, electrophysiological recordings, and behavioral assays.

Strengths:

(1) Non-synaptic interactions between neurons that share an extracellular environment (sometimes called "ephaptic" interactions) have not been well-studied, and certainly not in the insect taste system. A major strength of this study is the new insight it provides into how these interactions can impact sensory coding and behavior.

(2) The authors use many different types of genetic manipulations to dissect the role of HCN in GRN function, including mutants, RNAi, overexpression, ectopic expression, and neuronal silencing. Their results convincingly show that HCN impacts the sensillum potential and has both cell-autonomous and nonautonomous effects that go in opposite directions. Temporally controlled RNAi experiments suggest that the effect is not due to developmental changes. There are a couple of conflicting or counterintuitive results, but the authors discuss potential explanations.

(3) Experiments comparing flies raised on different food sources suggest an explanation for why the system may have evolved the way that it did: when flies live in a sugar-rich environment, their bitter sensitivity decreases, and HCN expression in sugar GRNs helps to counteract this decrease. New experiments in the revised paper show the timecourse of how sugar diet affects GRN responses and sensillum potential.

Weaknesses/Limitations:

(1) The RNAi Gal80ts experiment only compares responses of experimental flies housed at different temperatures without showing control flies (e.g. Gal4/+ and UAS/+ controls) to confirm that observed differences are not due to nonspecific effects of temperature. Certainly temperature cannot account for sugar and bitter GRN firing rates changing in opposite directions, but it may have some kind of effect.

(2) The experiments where flies are put on sugar vs. sorbitol food show that the diet clearly affects GRN responses and sensillum potential, even for food exposures as short as 1-4 hours, but it is not clear to what extent the GRNs in the labellum are being stimulated during those incubation periods. The flies are most likely not feeding over a 1 hour period if they were not starved beforehand, in which case it is not clear how many times the labellar GRNs would contact the food substrate.

(3) The authors mention that HCN may impact the resting potential in addition to changing the excitability of the cell through various mechanisms. It would be informative to record the resting potential and other neuronal properties, but this is very difficult for GRNs, so the current study is not able to determine exactly how HCN affects GRN activity.

---

## [Referee Report · Reviewer #2 (Public Review)]

Summary:

In this manuscript, the authors show that HCN loss-of-function mutation causes a decrease in spiking in bitter GRNs (bGRN) while leaving sweet GRN (sGRN) response in the same sensillum intact. They show that a perturbation of HCN channels in sweet-sensing neurons causes a similar decrease while increasing the response of sugar neurons. They were also able to rescue the response by exogenous expression. Ectopic expression of HCN in bitter neurons had no effect. Next, they measure the sensillum potential and find that sensillum potential is also affected by HCN channel perturbation. These findings lead them to speculate that HCN in sGRN increases sGRN spiking, which in turn affects bGRNs. To test this idea, they carried out multiple perturbations aimed at decreasing sGRN activity. They found that reducing sGRN activity by either using receptor mutant or by expressing Kir (a K+ channel) in sGRN increased bGRN responses. These responses also increase the sensillum potential. Finally, they show that these changes are behaviorally relevant as conditions that increase sGRN activity decrease avoidance of bitter substances.

Strengths:

There is solid evidence that perturbation of sweet GRNs affects bitter GRN in the same sensillum. The measurement of transsynaptic potential and how it changes is also interesting and supports the author's conclusion

Weaknesses:

The ionic basis of how perturbation in GRN affects the transepithelial potential, which in turn affects the second neuron, is unclear.

---

## [Referee Report · Reviewer #3 (Public Review)]

Ephaptic inhibition between neurons housed in the same sensilla has been long discovered in flies, but the molecular basis underlying this inhibition is underexplored. Specifically, it remains poorly understood which receptors or channels are important for maintaining the transepithelial potential between the sensillum lymph and the hemolymph (known as the sensillum potential), and how this affects the excitability of neurons housed in the same sensilla.

Lee et al. used single-sensillum recordings (SSR) of the labellar taste sensilla to demonstrate that the HCN channel, Ih, is critical for maintaining sensillum potential in flies. Ih is expressed in sugar-sensing GRNs (sGRNs) but affects the excitability of both the sGRNs and the bitter-sensing GRNs (bGRNs) in the same sensilla. Ih mutant flies have decreased sensillum potential, and bGRNs of Ih mutant flies have a decreased response to the bitter compound caffeine. Interestingly, ectopic expression of Ih in bGRNs also increases sGRN response to sucrose, suggesting that Ih-dependent increase in sensillum potential is not specific to Ih expressed in sGRNs. The authors further demonstrated, using both SSR and behavior assays, that exposure to sugars in the food substrate is important for the Ih-dependent sensitization of bGRNs. The experiments conducted in this paper are of interest to the chemosensory field. The observation that Ih is important for the activity in bGRNs albeit expressed in sGRNs is especially fascinating and highlights the importance of non-synaptic interactions in the taste system.

Comments on the revised version:

The authors performed additional analyses/experiments to address my previous major points. I'm satisfied with most of their answers:

(1) Sensilla types are labeled in all figures. Proper GAL4 and UAS controls were added to the figures.

(2) Fig. 2A was added to illustrate the important concepts of SP. Fig. 5E was added to show a working model, which could be better but is alright.

(3) Although not in my list of major points, I appreciate the newly added Fig. 5A and 5B, which demonstrate the long-lasting effect of exposure to sugars.

(4) Post-stimulus histogram was added for Fig. 4.

(5) Regarding the expression of Ih in bGRNs and sGRNs, the authors referred to their preprint (Lee et al., 2023, Fig 5C, D, suppl movie 1 and 2). The authors stated that "On the other hand, bGRNs labeled by Gr66a-LexA appeared to colocalize only partially with GFP when the confocal stacks were examined image by image." This interpretation unfortunately does not align with my viewing of the images and the movies. Just looking at the images and the movies alone, one would conclude that Ih is indeed expressed in both bGRNs and sGRNs. Notably, the Ih-TG4.0 is expressed in other non-neuronal cells in the labellum. That being said, I agree with the authors that even if Ih is indeed expressed in bGRNs, it would not affect SP (Fig. 1C, D of this paper, Fig. 5B of Lee et al., 2023 preprint), so I think the authors have addressed my major concern.

---

## [Author Response]

The following is the authors’ response to the original reviews.

**Reviewer #1 (Public Review):**
Summary:This study identifies new types of interactions between *Drosophila* gustatory receptor neurons (GRNs) and shows that these interactions influence sensory responses and behavior. The authors find that HCN, a hyperpolarization-activated cation channel, suppresses the activity of GRNs in which it is expressed, preventing those GRNs from depleting the sensillum potential, and thereby promoting the activity of neighboring GRNs in the same sensilla. HCN is expressed in sugar GRNs, so HCN dampens the excitation of sugar GRNs and promotes the excitation of bitter GRNs. Impairing HCN expression in sugar GRNs depletes the sensillum potential and decreases bitter responses, especially when flies are fed on a sugar-rich diet, and this leads to decreased bitter aversion in a feeding assay. The authors' conclusions are supported by genetic manipulations, electrophysiological recordings, and behavioral assays.Strengths:(1) Non-synaptic interactions between neurons that share an extracellular environment (sometimes called "ephaptic" interactions) have not been well-studied, and certainly not in the insect taste system. A major strength of this study is the new insight it provides into how these interactions can impact sensory coding and behavior.

We appreciate the reviewer’ view that our findings may allow researchers to better understand sensory coding and behavior. However, we respectfully disagree that the SP homeostasis in *Drosophila* gustation we describe here pertains to ephaptic interaction. Although SP reduction was proposed as the basis of post-ephaptic hyperpolarization in Drosophila olfaction, we find that SP changes are too slow to mediate the fast action of ephaptic inhibition in gustation, reported in the ref#17. We observed a slow, sweet-dependent SP depletion (Fig. 5B, revised), which takes more than one hour. The real-time change of SP was also slow even upon contact with 200-mM sucrose; this result was set aside for another manuscript in preparation. Therefore, we believe the main findings in this paper concern the homeostatic preservation of SP for the maintenance of gustatory function, not ephaptic interaction.

(2) The authors use many different types of genetic manipulations to dissect the role of HCN in GRN function, including mutants, RNAi, overexpression, ectopic expression, and neuronal silencing. Their results convincingly show that HCN impacts the sensillum potential and has both cell-autonomous and nonautonomous effects that go in opposite directions. There are a couple of conflicting or counterintuitive results, but the authors discuss potential explanations.(3) Experiments comparing flies raised on different food sources suggest an explanation for why the system may have evolved the way that it did: when flies live in a sugar-rich environment, their bitter sensitivity decreases, and HCN expression in sugar GRNs helps to counteract this decrease.Weaknesses/Limitations:(1) The genetic manipulations were constitutive (e.g. Ih mutations, RNAi, or misexpression), and depleting Ih from birth could lead to compensatory effects that change the function of the neurons or sensillum. Using tools to temporally control Ih expression could help to confirm the results of this study.

We attempted to address this point by using the tub-Gal80ts system. The result is now included as Fig. 1-figure supplement 2. At 29C, a non-permissive temperature for GAL80ts which allows GAL4-dependent expression Ih-RNAi, we observed that bGRN responses were decreased and sGRN responses were increased compared to the control maintained at 18°C, and this is in parallel with the result in Fig. 1C,D. For this experiment, we inserted “To exclude the possibility that Ih is required for normal gustatory development, we temporally controlled Ih RNAi knockdown to occur only in adulthood, which produced similar results (Fig. 1-figure supplement 2).” (~line 113).

(2) The behavioral experiment shows a striking loss of bitter sensitivity, but it was only conducted for one bitter compound at one concentration. It is not clear how general this effect is. The same is true for some of the bitter GRN electrophysiological experiments that only tested one compound and concentration.

We conducted additional behavioral experiments with other bitters such as lobeline and theophylline (Fig. 5-figure supplement 1), which showed sensitivity losses in Ih mutants similar to caffeine. For these results, the following is inserted at ~line 274: “These results were recapitulated with other bitters, lobeline and theophylline (Fig. 5-figure supplement 1).”

We also added single sensillum recording data with bitters, berberine, lobeline, theophylline and umbelliferone, which yielded results similar to those obtained with caffeine (Fig. 1-figure supplement 1). This is described with the sentence at ~line 105 “Other bitter chemical compounds, berberine, lobeline, theophylline, and umbelliferone, also required Ih for normal bGRN responses (Fig. 1-figure supplement 1).”

(3) Several experiments using the Gal4/UAS system only show the Gal4/+ control and not the UAS/+ control (or occasionally neither control). Since some of the measurements in control flies seem to vary (e.g., spiking rate), it is important to compare the experimental flies to both controls to ensure that any observed effects are in fact due to the transgene expression.

We appreciate the reviewers for raising this point. Indeed, there was a small logical flaw with the controls. We have now included all the necessary controls for Fig. 1C-F, Fig. 2I,J, Fig. 4E, and Fig. 5D, as reviewers suggested. These experiments remained statistically significant after including the new control groups.

(4) I was surprised that manipulations of sugar GRNs (e.g. Ih knockdown, Gr64a-f deletion, or Kir silencing) can impact the sensillum potential and bitter GRN responses even in experiments where no sugar was presented.

We are afraid there is a misunderstanding on the early part of the paper. We suspected that the manipulations impacted bGRNs and SP due to the sweetness in the regular cornmeal food, as stated in lines 214-220 “Typically, we performed extracellular recordings on flies 4-5 days after eclosion, during which they were kept in a vial with fresh regular cornmeal food containing ~400 mM D-glucose. The presence of sweetness in the food would impose long-term stimulation of sGRNs, potentially requiring the delimitation of sGRN excitability for the homeostatic maintenance of gustatory functions. To investigate this possibility, we fed WT and Ihf03355 flies overnight with either non-sweet sorbitol alone (200 mM) or a sweet mixture of sorbitol (200 mM) + sucrose (100 mM).”

I believe the authors are suggesting that the effects of sugar GRN activity (e.g., from consuming sugar in the fly food prior to the experiment) can have long-lasting effects, but it wasn't entirely clear if this is their primary explanation or on what timescale those long-lasting effects would occur. How much / how long of a sugar exposure do the flies need for these effects to be triggered, and how long do those effects last once sugar is removed?

We attempted to address this point with additional experiments (Fig. 5A,B). The reduction of SP could be observed in WT and HCN-deficient mutants with similar degrees 1 hr after the flies were transferred from nonsweet sorbitol-containing vials to sweet sucrose-containing ones. Moreover, the mutants, but not WT, showed further depression of SP when the sweetness persisted in the media for 4 hrs and overnight. This long-term exposure to sweetness longer than 1 hr may simulates the feeding on the regular sweet cornmeal food. The recovery of SP was also tested by removing flies from the sweet media after overnight-long sweet exposure and placing them in sorbitol food. SPs of WT and the mutants were recovered to the similar levels 1 hr after separating the animals from sweetness, although the HCN-lacking mutants showed much lower SP right after overnight sweetness exposure. The unimpaired recovery of the mutants suggests that HCN is independent of generating transepithelial potential itself. Therefore, regardless of HCN, SP changes are not fast even in the presence of strong sweetness, and SP is much better guarded when sGRNs express HCN in a sweet environment.

We inserted the following at ~line 260 to describe the newly added recovery experiment: “Following overnight sweet exposure, SPs of WT and Ihf03355 were recovered to similar levels after 1-hr incubation with sorbitol only food. However, it was after 4 hrs on the sorbitol food that the two lines exhibited SP levels similar to those achieved by overnight incubation with sorbitol only food (Fig. 5B). These results indicate that SP depletion by sweetness is a slow process, and that the dysregulated reduction and recovery of SPs in Ihf03355 manifest only after long-term conditioning with and without sweetness, respectively.”.

(5) The authors mention that HCN may impact the resting potential in addition to changing the excitability of the cell through various mechanisms. It would be informative to record the resting potential and other neuronal properties, but this is very difficult for GRNs, so the current study is not able to determine exactly how HCN affects GRN activity.

On this point, we cannot but rely on previous studies of biophysical and electrophysiological characterization on mammalian HCN channels and a heterologous expression study that revealed a robust hyperpolarization-activated cation current from *Drosophila* HCN channels (PMID: 15804582).

**Reviewer #2 (Public Review):**
Summary:In this manuscript, the authors start by showing that HCN loss-of-function mutation causes a decrease in spiking in bitter GRNs (bGRN) while leaving sweet GRN (sGRN) response in the same sensillum intact. They show that a perturbation of HCN channels in sweet-sensing neurons causes a similar decrease while increasing the response of sugar neurons. They were also able to rescue the response by exogenous expression. Ectopic expression of HCN in bitter neurons had no effect. Next, they measure the sensillum potential and find that sensillum potential is also affected by HCN channel perturbation. These findings lead them to speculate that HCN in sGRN increases sGRN spiking which in turn affects bGRNs. To test this idea that carried out multiple perturbations aimed at decreasing sGRN activity. They found that decreasing sGRN activity by either using receptor mutant or by expressing Kir (a K+ channel) in sGRN increased bGRN responses. These responses also increase the sensillum potential. Finally, they show that these changes are behaviorally relevant as conditions that increase sGRN activity decrease avoidance of bitter substances.Strengths:There is solid evidence that perturbation of sweet GRNs affects bitter GRN in the same sensillum. The measurement of transsynaptic potential and how it changes is also interesting and supports the authors' conclusion.Weaknesses:The ionic basis of how perturbation in GRN affects the transepithelial potential which in turn affects the second neuron is not clear.

We speculate that HCN-dependent membrane potential regulation, rather than ionic composition change, is responsible for the observed SP preservation, as further discussed as an author response in the section of “Recommendations for the authors”. The transepithelial potential can be dissipated by increased conductance through receptor-linked ion channels following gustatory receptor activation in GRNs. The volume of the sensillum lymph is very small according to electron micrographs of horizontally sliced bristles (PMID: 11456419). Therefore, robust excitation of a gustatory neuron may easily deplete the extracellular potential built as a form of polarized ion concentrations across the tight junction. When the consumption is too strong and extended, the neighboring neuron, which share TEP with the activated GRN, can be negatively affected. We propose that HCN suppresses overexcitation of sGRNs by means of membrane potential stabilization. This stabilization prevents sGRNs from excessively reducing the TEP, thereby protecting the activity of neighboring bGRNs.

**Reviewer #3 (Public Review):**
Ephaptic inhibition between neurons housed in the same sensilla has been long discovered in flies, but the molecular basis underlying this inhibition is underexplored. Specifically, it remains poorly understood which receptors or channels are important for maintaining the transepithelial potential between the sensillum lymph and the hemolymph (known as the sensillum potential), and how this affects the excitability of neurons housed in the same sensilla.

Although a reduction of sensillum potential was proposed to underlie membrane hyperpolarization of post-ephaptic olfactory neurons in *Drosophila*, our preliminary data (not shown due to a manuscript in preparation) and the results included in the paper (Fig. 5B) strongly suggest that SP reduction is not a requisite for ephaptic inhibition at least in GRNs. Ephaptic inhibition is expected to be instantaneous, whereas we find that SP reduction in gustation is very slow. Therefore, we would like to indicate that the findings we report in this manuscript are not directly related to ephaptic inhibition.

Lee et al. used single-sensillum recordings (SSR) of the labellar taste sensilla to demonstrate that the HCN channel, Ih, is critical for maintaining sensillum potential in flies. Ih is expressed in sugar-sensing GRNs (sGRNs) but affects the excitability of both the sGRNs and the bitter-sensing GRNs (bGRNs) in the same sensilla. Ih mutant flies have decreased sensillum potential, and bGRNs of Ih mutant flies have a decreased response to the bitter compound caffeine. Interestingly, ectopic expression of Ih in bGRNs also increases sGRN response to sucrose, suggesting that Ih-dependent increase in sensillum potential is not specific to Ih expressed in sGRNs. The authors further demonstrated, using both SSR and behavior assays, that exposure to sugars in the food substrate is important for the Ih-dependent sensitization of bGRNs. The experiments conducted in this paper are of interest to the chemosensory field. The observation that Ih is important for the activity in bGRNs albeit expressed in sGRNs is especially fascinating and highlights the importance of non-synaptic interactions in the taste system.Despite the interesting results, this paper is not written in a clear and easily understandable manner. It uses poorly defined terms without much elaboration, contains sentences that are borderline unreadable even for those in the narrower chemosensory field, and many figures can clearly benefit from more labeling and explanation. It certainly needs a bit of work.

We would like to revise the language aspect of the manuscript after finalizing the scientific revision.

Below are the major points:(1) Throughout the paper, it is assumed that Ih channels are expressed in sugar-sensing GRNs but not bitter-sensing GRNs. However, both this paper and citation #17, another paper from the same lab, contain only circumstantial evidence for the expression of Ih channels in sGRNs. A simple co-expression analysis, using the Ih-T2A-GAL4 line and Gr5a-LexA/Gr66a-LexA line, all of which are available, could easily demonstrate the co-expression. Including such a figure would significantly strengthen the conclusion of this paper.

We did conduct confocal imaging with Ih-T2A-Gal4 in combination with GRN Gal4s (ref#17 version2). The expression is very broad, including both neurons and non-neuronal cells. We observed much stronger sGRN expression than bGRN expression. But the promiscuous expression of the reporter in many cells hindered us from clearly demonstrating the void of the reporter in bGRNs. However, the functional and physiological examination of Ih-T2A-Gal4 with the neuronal modifiers such as TRPA1 and Kir2.1 in ref#17 indicates the strong and little expression of Ih in sGRNs and bGRNs, respectively. Furthermore, the RNAi kd results present another line of evidence that HCN expressed in sGRNs regulates SP and bGRN activity (Fig. 1C,D, Fig. 1-figure supplement 2). Ih-RNAi expression in bGRNs did not result in any statistically significant changes in the activities of sGRNs and bGRNs compared to controls (Fig. 1C,D, revised), advocating that Ih acts in sGRNs for the functional homeostasis of SP and GRNs, as we claim.

(2) Throughout this paper, it is often unclear which class of labellar taste sensilla is being recorded. S-a, S-b, I-a, and I-b sensilla all have different sensitivities to bitters and sugars. Each figure should clearly indicate which sensilla is being recorded. Justification should be provided if recordings from different classes of sensilla are being pooled together for statistics.

We mainly performed SSR (single sensillum recording) on i-type bristles as they have the simplest composition of GRNs compared to s- and L-type bristles. As single s-types also contain each of s- and bGRN, we measured SP also for s-types (Figs. 2, 3F and 4D). In case of Fig.3-figure supplement 1, L-types were tested for the relationship between water cell activity and SP. Now all the panels are labelled with the tested bristle types.

(3) In many figures, there is a lack of critical control experiments. Examples include Figures 1C-F (lacking UAS control), Figure 2I-J (lacking UAS control), Figure 4E (lacking the UAS and GAL4 control, and it is also strange to compare Gr64f > RNAi with Gr66a > RNAi, instead of with parental GAL4 and UAS controls.), and Figure 5D (lacking UAS control). Without these critical control experiments, it is difficult to evaluate the quality of the work.

Thank you for pointing this out. We appreciate the feedback and have addressed these concerns by including all the requested controls in the figures. Specifically, we have added the UAS controls for Figs 1C-F and 2I-J, as well as the UAS and GAL4 controls for Fig. 4E. We have also included the UAS control for Fig. 5D.

(4) Figure 2A could benefit from more clarification about what exactly is being recorded here. The text is confusing: a considerable amount of text is spent on explaining the technical details of how SP is recorded, but very little text about what SP represents, which is critical for the readers. The authors should clarify in the text that SP is measuring the potential between the sensillar lymph, where the dendrites of GRNs are immersed, and the hemolymph. Adding a schematic figure to show that SP represents the potential between the sensillar lymph and hemolymph would be beneficial.

SP was defined at lines 55-56 in the first paragraph of introduction, which also contains the background information for SP as a transepithelial potential. As reviewer suggested, we now also included a sentence describing SP (“SP is known as a transepithelial potential between the sensillum lymph and the hemolymph, generated by active ion transport through support cells”, line 126) and a drawing to illustrate the concept of SP (Fig. 2A), and revised the legend.

(5) The sGRN spiking rate in Figure 4B deviates significantly from previous literature (Wang, Carlson, eLife 2022; Jiao, Montell PNAS 2007, as examples), and the response to sucrose in the control flies is not dosage-dependent, which raises questions about the quality of the data. Why are the responses to sucrose not dosage-dependent? The responses are clearly not saturated at these (10 mM to 100 mM) concentrations.

Our recordings show different spiking frequencies from others’ work, because the frequencies are from 5-sec bins not only first 0.5 sec. This lowers the frequencies, as spikes are relatively more frequent in the beginning of the recording (Fig. 4-figure supplement 1).

Why are the responses to sucrose not dosage-dependent? The responses are clearly not saturated at these (10 mM to 100 mM) concentrations.

We were also puzzled with the flat dose dependence to sucrose. This result may suggest the existence of another mechanism moderating sucrose responses of sGRNs. This flat curve reappeared with other genotypes with the same concentration range (5-50 mM) in Fig. 4E. However, 1-mM sucrose produced much lower spiking frequencies (Fig. 4E), suggesting that sGRN responses are saturated at 5 mM sucrose with our recording/analysis condition.

(6) In Figure 4C, instead of showing the average spike rate of the first five seconds and the next 5 seconds, why not show a peristimulus time histogram? It would help the readers tremendously, and it would also show how quickly the spike rate adapts to overexpression and control flies. Also, since taste responses adapt rather quickly, a 500 ms or 1 s bin would be more appropriate than a 5-second bin.

Taste single sensillum recording starts by contacting stimulants, which bars us from recording pre-stimulus responses of GRNs. Therefore, we showed post-stimulus graphs with 1-sec bins (Fig. 4-figure supplement 1) as we reviewer suggested.

(7) Lines 215 - 220. The authors state that the presence of sugars in the culture media would expose the GRNs to sugar constantly, without providing much evidence. What is the evidence that the GRNs are being activated constantly in flies raised with culture media containing sugars? The sensilla are not always in contact with the food.

We agree with reviewer. We replaced “long-term stimulation of sGRNs” with “strong and frequent stimulation of sGRNs for extended period”. The word long-term may be interpreted to be constant.

(8) Line 223. To show that bGRN spike rates in Ih mutant flies "decreased even more than WT", you need to compare the difference in spike rates between the sorbitol group and the sorbitol + sucrose group, which is not what is currently shown.

The data were examined by ANOVA and a multiple comparison test (Dunn’s) between all the groups regardless of genotypes and conditions in the panel (all the groups sharing the y axis). Therefore, the differences were statistically examined. However, the cited expression we used read like it was about the slope or extent of the decrease. We intended to indicate the difference in the absolute values of spiking frequencies after overnight sweet exposure between the genotypes, while bGRN activities were statistically indifferent between WT and Ih mutants when they were kept only on sorbitol food. We revised it to “decreased to the level significantly lower than WT”. We also changed the graph style to effectively present the trend of changes in bGRN sensitivity with comparison between genotypes. Again, the groups were statistically examined together regardless of the genotypes and conditions.

(9) To help readers better understand the proposed mechanisms here, including a schematic figure would be helpful. This should show where Ih is expressed, how Ih in sGRNs impacts the sensillum potential, how elevated sensillum potential increases the electrical driving force for the receptor current, and affects the excitability of the bGRNs in the same sensilla, and how exposure to sugar is proposed to affect ion homeostasis in the sensillum lymph.

As reviewer suggested, we included two panels to show working model for gustatory homeostasis via SP maintenance by HCN (Fig. 5E,F).

**Reviewer #1 (Recommendations For The Authors):**
(1) The relationship between this paper and the authors' bioRxiv preprint posted last year is not clear. In the introduction they made it seem like this paper is a follow-up that builds on the preprint, but most or all of the experiments in this paper were already performed in the preprint. I guess the authors are planning to divide the original paper into two papers. I would suggest updating the preprint to avoid confusion.

Thank you for the comment. We updated the preprint to be without a part of Fig.6 and entire Fig.7 along with associated texts. As reviewer pointed out, our eLife paper was spun off from the part of the preprint paper, because we feel that the two stories could confuse readers when presented together.

(2) Have the authors considered testing responses of water GRNs? They reside in the same sensilla as sugar neurons, so are they also increased affected by Ih mutation or RNAi in sugar neurons? This would strengthen the evidence that the indirect (non-cell autonomous) effects of Ih are due to the sensillum potential and not some specific interaction between sweet and bitter cells.

As reviewer proposed, we appraised water GRN activity in the L-type bristles of WT, Ihf03355 and a genomic rescue line for Ihf03355. Spiking responses in water GRNs were evoked by hypo-osmolarity of electrolyte (0.1 mM tricholine citrate-TCC). Interestingly, the Ih mutant showed reduced 0.1 mM TCC-provoked spiking frequencies compared to WT. This impairment was rescued by the genomic fragment containing an intact Ih locus (Figure 3-figure supplement 1A).

Additionally, SPs in L-type bristles were reduced by Ih deficiencies but increased in Gr64af, suggesting that HCN regulates sGRNs in L-type bristles as well (Figure 3-figure supplement 1B). Again, the bristles of animals with both mutations together exhibited SPs similar to those of WT.

Furthermore, when we conducted cDNA rescue experiments in L bristles, introduction of Ih-RF cDNA in sGRNs restored SPs, while expressing it in bGRNs did not unlike the results from the i- and s-bristles (Fig. 2K,L), likely because L-bristles lack bGRNs. These cDNA rescue and genetic interaction experiments were conducted using flies fed on fresh cornmeal food with strong sweetness, suggesting that the sweetness in the media is the likely key factor producing the genetic interaction and necessitating HCN, consistent with other results in the manuscript. Therefore, SP regulation by HCN is observed in the L-type bristles.

Minor comments:Line 52: typo, "Many of"

Thank you. Corrected

Line 95: typo, "sensilla do an sGRN"

Corrected

Line 98: typo, "we observed reduced the spiking responses"

Corrected

Line 206: typo, "a relatively low sucrose concentrations"

Corrected

Line 260: "inverse relationship between the two GRNs in excitability" - I am not exactly sure what data you are referring to.

Although alleles did not show increased sGRN activities, knockdown of Ih decreased bGRN activity but increased sGRN activity (Fig. 1C,D, Fig.1-figure supplement 2B), while suppression of sGRNs increased bGRN activity (Fig. 3). To clarify this point, we revised the phrase to “the inverse relationship between the two GRNs in excitability observed in Fig. 1C,D, Fig. 1-figure supplement 2B, and Fig. 3”.

Methods: typo, "twenty of 3-5 days with 10 males and 10 females"

Corrected to “Twenty flies, aged 3-5 days and consisting of 10 males and 10 females,”

Methods: typo, "Kim's wipes" should be "Kimwipes"

Corrected

**Reviewer #2 (Recommendations For The Authors):**
(1) More clarification is necessary on Transepithelial potential (TEP). TEP is typically created by having pumps and tight junctions between the sensillar lymph and the hemolymph.

We have an introduction to TEP or SP in the context of sensory functions (lines 40-57) with relevant references. The involvement of pumps and tight junction was mentioned in the same paragraph; “Glia-like support cells exhibit close physical association with sensory receptor neurons, and conduct active transcellular ion transport, which is important for the operation of sensory systems” (line 40) and “Tight junctions between support cells separate the externally facing sensillar lymph from the internal body fluid known as hemolymph” (line 53).

It is not clear how HCN channels in one of the neurons might change the composition of the sensillum lymph. An explanation of their model of how TEP depends on HCN is necessary.

Although the ionic composition of the sensillum lymph is a contributing factor to the sensillum potential, it is more conceptually relevant to describe our findings with the perspective of membrane potential regulation given the role of HCN in membrane potential stabilization as discussed in our manuscript.

We speculate that HCN controls the membrane potential at rest and/or in motion to modulate sGRN activity towards saving SP despite the sweetness in the niche. We positioned our results in relation to SP in discussion; “Our results provide multiple lines of evidence that HCN suppresses HCN-expressing GRNs, thereby sustaining the activity of neighboring GRNs within the same sensilla. We propose that this modulation occurs by restricting SP consumption through HCN-dependent neuronal suppression rather than via chemical and electrical synaptic transmission.” (lines 252-255). Moreover, it is unclear whether HCN is localized to the dendrite bathed in the sensillum lymph to influence the ionic composition of the lymph. It would be very interesting to study in future whether the ionic flow through HCN channels itself is critical for the function of HCN in this context, and whether HCN is exclusively present in the dendrite to support the postulation. However, we would like to remind reviewer that Kir2.1 and HCN channels in sGRNs showed similar effects on SP and bGRNs, while they differ in Na+ conductance.

In the initially submitted manuscript (lines 325-343), we discussed the potential mechanism by which Kir2.1 and HCN channels commonly increase SP in terms of how the membrane potential regulation in the soma can control the SP consumption in the dendrite of sGRNs.

Another point about the TEP that needs some explanation is that these sensilla are open to the environment as tastants must flow in and are different from mechanical sensilla in that sense.

This is a very important question regarding the general physiology of the taste sensilla, as the sensillum lymph is in contact with the external environment through the pore of the sensillum. It is indeed interesting to consider how the composition and potential of the lymph are maintained despite the relatively vast volume of food the sensilla encounter during gustation and the continuous evaporation to air between episodes of gustation. However, we believe that this question, while important, is distinct from the primary focus of our manuscript.

Are the TEP measurements in Figure 2 under control conditions where there are no tastants?

There is no tastant in the SP-measuring glass electrode other than the electrolyte. We apologize that we did not specify the recording electrode condition. We inserted a clause in the method; “For SP recordings, the recording electrode contained 2 mM TCC as the electrolyte, and…”

Does the TEP change dynamically as sGRN is activated?

SP does shift in response to sweets. Please see Fig. 5B. Also, we showed SP changes by mechanical stimuli, which depended on the mechanoreceptor, NompC (Fig. 2D-F). Mechanoreceptor neurons share the sensillum lymph with GRNs.

(2) More clarification on the potential transduction mechanism and how TEP affects one neuron differentially. Essentially, sGRN perturbation affects sGRN activity and it affects the TEP. More explanation is needed for the potential ionic mechanism of each.

Our results strongly suggest that HCN lowers the activity of HCN-expressing GRNs, mitigating SP consumption. This modulation is crucial because the SP serves as a driving force for neuronal activation within the sensillum. HCN is particularly necessary in sGRNs because of the flies’ sweet feeding niche, which is expected to result in frequent and strong activation of sGRNs. The SP saved by HCN-dependent delimitation of sGRNs can be used to raise the responsibility of bGRNs.

(3) The authors refer to their own unreviewed paper (Reference 17). This paper is on a similar topic and there seems to be some overlap. Clarification on this point would be important.

We revised the biorxiv preprint, so that the preprint version 2 does not contain the parts overlapping with this eLife paper. This eLife paper was originally part of the preprint paper, but it was separated to clarify the messages of the two stories. As we explained in Discussion (lines 276-297), HCN provides resistance to both hyperpolarization and depolarization of the membrane potential. Simply put, one paper focuses on the role of HCN in resisting hyperpolarization, while the other (this paper in eLife) focuses on resisting depolarization.

(4) Methods are sparse. Many details on the method are necessary. For example, Sensilla recordings are being done by the tip-dip method (I assume). What does "number of experiments" mean in Figure 1? Is it the number of animals or the number of sensilla? How many trials/sensilla?

We indicated the extracellular recording was performed by the tip-dip method; “In vivo extracellular recordings were performed by the tip-dip method as detailed previously”. We also added a statement on the number of experiments; “The number of experiments indicated in figures are the number of naïve bristles tested. The naïve bristles were from at least three different animals.”

(5) Figure 1: I understand the author's interpretation. But if one compares WT in Figure 1A to Gr64a-IhRNAi in 1C, we can come to the conclusion that there is no change. In other words, the control in Figure 1C (grey) has a much higher response than WT. Similar conclusions can be made for other experiments. Is the WT response stable enough to make the conclusions made here?

The genetic background of each genotype may influence GRN activity to some extent. RNAi knockdown experiments are well-known for their hypomorphic nature, and their effects should be evaluated by comparison with their parental controls such as Gal4 and UAS lines. As all reviewers pointed out, we added the results from UAS control. This effort confirms that Gr89a>Ih RNAi is statistically indifferent to UAS control as well as Gr64f-Gal4 control in bGRN spiking evoked by 2-mM caffeine, while Gr64f>Ih RNAi showed reduced bGRN responses to 2 mM caffeine compared to all the controls.

(6) Figure 3: Why is bGRN spiking not plotted against sensillum potential to observe the dependence more directly?

This is a very interesting suggestion. We are not, however, equipped to measure spiking and sensillum potential simultaneously. Therefore, they are independent experiments, and we treated them accordingly.

(7) Figure 4: Why bGRN response is only affected at high caffeine concentrations is not clear.

We were also surprised by the differences in the dose dependence results of b- and sGRNs, genetically manipulated to mis-express and over-express HCN in Fig. 4A and 4E, respectively. Each gustatory neuron likely has distinct sets of players and parameters that set its own membrane potential and excitability.

We can think of a possibility that there might be a range of membrane potentials within which HCN does not engage. In bGRNs, the resting membrane potential may lie low within this range, so that some degrees of membrane depolarization by low concentrations of caffeine do not significantly close HCN channels, thus preventing their hyperpolarizing effects. On the other hand, the membrane potential of sGRNs may be high within this range, showing suppressive effects at all tested sucrose concentrations. However, we find this explanation is too speculative to include in the main text, while we stated in the original manuscript, “implying a complex cell-specific regulation of GRN excitability.” (line 210).

(8) Minor:L98 - there is a small typo

Corrected

L274: "funny" !?

“Funny” currents, denoted If, were initially observed by electrophysiologists and later attributed to HCN channels, now indicated by Ih (thus the gene name Ih in *Drosophila*). These currents were termed "funny" due to their unusual properties compared to other currents. For more detailed information, please refer to the cited references.

L257: Neuropeptide seemed to be abrupt

We attempted to discuss possible mechanisms that mediate excitability changes across GRNs beyond the mechanism by SP shifts. Neuropeptides, which are chemical neurotransmitters along with small neurotransmitters, were mentioned following the discussion on synaptic transmission to suggest alternative pathways for excitability regulation. This inclusion is meant to provide a comprehensive overview of potential mechanisms influencing GRN activity.

**Reviewer #3 (Recommendations For The Authors):**
Congratulations on your fascinating research! The results are certainly of interest to the chemosensory field. However, I suggest using academic editing services to enhance the clarity of your text and ensure that the terminology and jargon align with standard usage in the field. The current choice of words may not be consistent with commonly used terms. As it is now, the writing might not fully showcase the compelling story and the effort behind your study, and is underselling your interesting results. Proper refinement could make sure your valuable findings are appropriately recognized.

We appreciate your comments and apologize for any difficulties reviewers faced during the review process. We are currently prioritizing the review of scientific content and plan to address language issues in a subsequent revision. It would be very helpful for future revisions if the problematic sentences or expressions could be indicated in detail after this revision. This will allow us to ensure that our terminology and expression align with standard usage in the field, and that our findings are clearly and effectively communicated.

Minor points:(1) Line 110: what is Ih-RF?

We apologize that we relied on a reference in describing the cDNA. The following clause was inserted with additional reference and the Flybase id: “(Flybase id: FBtr0290109), which previously rescued Ih deficiency in other contexts17,26 ,”

(2) Line 158: Gr64af mutant flies still have Gr5a and a residual response to fructose and sucrose (Slone, Amrein 2007).

We revised the line to “is severely impaired in sucrose and glucose sensing”, since there is a substantial loss of sucrose and glucose sensing in both Gr64af from Kim et al 2018 and DGr64 from Slone et al 2007, when they were examined by the proboscis extension reflex assay. This was also confirmed in the study by Jiao et al 2009. We also deleted “sugar-ageusic” and instead describe the mutant “impaired in sucrose and glucose sensing” in Fig. 3 legend.

(3) Lines 264-273 seem unnecessary. This paper is not about the function of HCN in mammals, and these discussions seem largely irrelevant.

We feel that it is important to position our results within a broader context by discussing the potential implications of our findings for sensory systems of other animals. As we stated, HCN channels have been localized in mammalian sensory systems, but their roles are often not well understood. By including this discussion, we aim to highlight the relevance of our findings beyond the model organism used in our study and suggest possible areas for future research in mammalian systems.